# Representation Learning for Online and Offline RL in Low-rank MDPs

**Masatoshi Uehara**
Department of Computer Science
Cornell University, Ithaca, NY 14850, USA
mu223@cornell.edu

**Xuezhou Zhang**
Department of Electrical and Computer Engineering
Princeton University, NJ 08544,USA
xz7392@princeton.edu

**Wen Sun**
Department of Computer Science, Cornell University, Ithaca, NY 14850, USA
ws455@cornell.edu

## Abstract

This work studies the question of *Representation Learning in RL:* how can we learn a compact low-dimensional representation such that on top of the representation we can perform RL procedures such as exploration and exploitation, in a sample efficient manner. We focus on the low-rank Markov Decision Processes (MDPs) where the transition dynamics correspond to a low-rank transition matrix. Unlike prior works that assume the representation is known (e.g., linear MDPs), here we need to learn the representation for the low-rank MDP. We study both the online RL and offline RL settings. For the online setting, operating with the same computational oracles used in FLAMBE(Agarwal et al., 2020b)—-the state-of-art algorithm for learning representations in low-rank MDPs, we propose an algorithm REP-UCB—Upper Confidence Bound driven REPresentation learning for RL, which *significantly improves* the sample complexity from $\widetilde{O}(A^9 d^7/(\epsilon^{10}(1-\gamma)^{22}))$ for FLAMBE to $\widetilde{O}(d^4 A^2/(\epsilon^2(1-\gamma)^5))$ with $d$ being the rank of the transition matrix (or dimension of the ground truth representation), $A$ being the number of actions, and $\gamma$ being the discount factor. Notably, REP-UCB is simpler than FLAMBE, as it directly balances the interplay between representation learning, exploration, and exploitation, while FLAMBE is an explore-then-commit style approach and has to perform reward-free exploration step-by-step forward in time. For the offline RL setting, we develop an algorithm that leverages pessimism to learn under a partial coverage condition: our algorithm is able to compete against *any policy* as long as it is covered by the offline data distribution.

## 1 Introduction

When applying Reinforcement Learning (RL) to large-scale problems where data is complex and high-dimensional, learning effective transformations of the data, i.e., representation learning, can often significantly improve the sample and computation efficiency of the RL procedure. Indeed, several empirical works have shown that leveraging representation learning techniques developed in supervised or unsupervised learning settings can accelerate the search for good decision-making strategies (Silver et al., 2018; Stooke et al., 2021; Srinivas et al., 2020; Yang & Nachum, 2021). However, representation learning in RL is far more subtle than it is for non-sequential and non-interactive learning tasks (e.g., supervised learning). Prior works have shown that even if one is given the magic representation that exactly linearizes the optimal policy (Du et al., 2019b) or the optimal value functions (Wang et al., 2020; Weisz et al., 2021), RL is still challenging (i.e., one may still need exponentially many samples to learn). This indicates that an effective representation that permits efficient RL needs to encode more information about the underlying Markov Decision Processes (MDPs). Despite the recent empirical success of representation learning in RL , its statistical guarantee and theoretical properties remain under-investigated.

In this work, we study the representation learning question under the low-rank MDP assumption. Concretely, a low-rank MDP assumes that the MDP transition matrix admits a low-rank factorization, i.e., there exists two *unknown* mappings $\mu(s'), \phi(s,a)$, such that $P(s'|s,a) = \mu(s')^\top \phi(s,a)$ for all $s, a, s'$, where $P(s'|s,a)$ is the probability of transiting to the next state $s'$ under the current

state and action $(s, a)$. The representation $\phi$ in a low-rank MDP not only linearizes the optimal state-action value function of the MDP (Jin et al., 2020a), but also linearizes the transition operator. A low-rankness assumption on large stochastic matrices is a common and natural assumption and has enabled successful development of algorithms for real world applications such as movie recommendation systems (Koren et al., 2009). We note that a low-rank MDP strictly generalizes the linear MDP model (Yang & Wang, 2020; Jin et al., 2020a) which assumes $\phi$ is known a priori. The unknown representation $\phi$ makes learning in low-rank MDPs much more challenging than that in linear MDPs since one can no longer directly use linear function approximations. On the other hand, the fact that linear MDPs can be solved statistical and computational efficiently if $\phi$ is known a priori implies that if one could learn the representation of the low-rank MDP, one could then efficiently learn the optimal policy.

Indeed, prior works have shown that learning in low-rank MDPs is statistically feasible (Jiang et al., 2017; Sun et al., 2019; Du et al., 2021) via leveraging rich function approximators. However, these algorithms are version space algorithms and are not computationally efficient. Recent work FLAMBE proposes an oracle-efficient algorithm[1] that learns in low-rank MDPs with a polynomial sample complexity, where the computation oracle is Maximum Likelihood Estimation (MLE) operating under the standard supervised learning style Empirical Risk Minimization (ERM) setting. In this work, we follow the same setup from FLAMBE (Agarwal et al., 2020b), and propose a new algorithm — *Upper Confidence Bound driven Representation Learning, Exploration and Exploitation (*REP-UCB*)*, which can learn a near optimal policy for a low-rank MDP with a polynomial sample complexity and is oracle-efficient. Comparing to FLAMBE, our algorithm *significantly improves* the sample complexity from $O(d^7 A^9 / (\epsilon^{10}(1-\gamma)^{22}))$ for FLAMBE to $O(d^4 A^2 / (\epsilon^2(1-\gamma)^5))$, where $d$ is the rank of the transition matrix (or dimension of the true representation), $A$ is the number of actions, $\epsilon$ is the suboptimality gap and $\gamma \in [0, 1)$ is the discount factor in the MDP. Our algorithm is also arguably much simpler than FLAMBE: FLAMBE is an explore-then-commit algorithm, has to explore in a layer-by-layer forward way, and does not permit data sharing across different time steps. In contrast, REP-UCB carefully trades exploration versus exploitation by combining the reward signal and exploration bonus (constructed using the latest learned representation), and enables data sharing across all time steps.[2] Our sample complexity nearly matches the ones from those computationally inefficient algorithms (Jiang et al., 2017; Sun et al., 2019; Du et al., 2021). We summarize the comparison with the prior works that study representation learning in Table 1.

In addition to the online exploration setting, we also show that our new techniques can be directly used for designing offline RL algorithms for low-rank MDPs under partial coverage. More specifically, we propose an algorithm REP-LCB—*Lower Confidence Bound driven Reprepresentation Learning for offline RL*, that given an offline dataset, can learn to compete against any policy (including history-dependent policies) as long as it is covered by the offline data where the coverage is measured using the relative condition number (Agarwal et al., 2021) associated with the ground truth representation. Thus, our offline RL result generalizes prior offline RL works on linear MDPs (Jin et al., 2020b; Zhang et al., 2021b) which assume representation is known a priori and use linear function approximation. Computation-wise, our approach uses one call to the MLE computation oracle, and hence is oracle-efficient. REP-LCB is *the first oracle efficient offline algorithm for low-rank MDP enjoying the aforementioned statistical guarantee.* See Section 2 for a more detailed comparison with the existing literature on representation learning in offline RL.

**Our contributions.** We develop new representation learning RL algorithms that enable sample efficient learning in low-rank MDPs under both online and offline settings:

1. In the online episodic learning setting, our new algorithm REP-UCB integrates representation learning, exploration, and exploitation together, and significantly improves the sample complexity of the prior state-of-art algorithm FLAMBE;

---

[1]The oracle generally refers to supervised learning style empirical risk minimization oracle. We seek to design an algorithm that runs in polynomial time with each oracle call counting as $O(1)$. The reduction to supervised learning has lead to many successful provable and practical algorithms in contextual bandit (Agarwal et al., 2014; Dudík et al., 2017; Foster & Rakhlin, 2020) and RL (Du et al., 2019a; Misra et al., 2020).

[2]Our algorithm and analysis can be easily extended to finite horizon non-stationary setting. We choose the discounted infinite horizon setting to contrast our results to FLAMBE: FLAMBE is *not* capable of learning stationary policies under the discounted infinite horizon setting.

| Methods | Setting | Sample Complexity | Computation |
|---------|---------|-------------------|-------------|
| OLIVE (Jiang et al., 2017) | Low Bellman rank | $\frac{d^2 A}{\epsilon^2(1-\gamma)^4}$ | Inefficient |
| Witness rank (Sun et al., 2019) | Low Witness rank | $\frac{d^2 A}{\epsilon^2(1-\gamma)^4}$ | Inefficient |
| BLin-UCB (Du et al., 2021) | Bilinear Class | $\frac{d^2 A}{\epsilon^2(1-\gamma)^7}$ | Inefficient |
| Moffle (Modi et al., 2021) | Low-nonnegative-rank MDP | $\frac{d^6 A^{13}}{\epsilon^2 \eta^5 (1-\gamma)^5}$ | Oracle efficient |
| FLAMBE Agarwal et al. (2020b) | Low-rank MDP | $\frac{d^7 A^9}{\epsilon^{10}(1-\gamma)^{22}}$ | Oracle efficient |
| REP-UCB (Ours) | Low-rank MDP | $\frac{d^4 A^2}{\epsilon^2(1-\gamma)^5}$ | Oracle efficient |

Table 1: Comparison among different provable representation learning algorithms in online RL. Algorithms such as OLIVE, Witness rank, and BLin-UCB work for settings which are more general than low-rank MDPs and have tight sample complexity. However, these algorithms are version space algorithms and thus are not computationally efficient. Moffle is an oracle-efficient algorithm (with a much stronger oracle than the one in FLAMBE and ours), but the assumptions under which Moffle operates essentially imply that the MDP's transition has low non-negative matrix rank (nnr). Note that a nnr is at least as large as and could be exponentially larger than the rank (Agarwal et al., 2020b). Finally, FLAMBE operates under the same function approximation setting and the computation oracle as ours. Our algorithm *significantly improves* the sample complexity from FLAMBE in *all parameters*. Note the horizon dependence is not exactly comparable as these prior works originally considered the finite horizon setting with nonstationary transition, and we convert their results to the discounted setting by simply replacing the finite horizon H by $\Theta(1/(1-\gamma))$.

2. In the offline learning setting, we propose a natural concentrability coefficient (i.e., relative condition number under the true representation) that captures the partial coverage condition in low-rank MDP, and our algorithm REP-LCB learns to compete against any policy (including history-dependent ones) under such a partial coverage condition.

## 2 RELATED WORK

**Online Setting** We list the comparison as follows, which is summarized in Table 1. Additional related works are discussed in Section A.

FLAMBE (Agarwal et al., 2020b) was a state-of-the-art oracle-efficient algorithm for low-rank MDPs. In all parameters, the statistical complexity is much worse than REP-UCB . Our algorithm and FLAMBE operate under the same computation oracle. FLAMBE does not balance exploration and exploitation, and uses explore-then-committee style techniques (i.e., constructions of absorbing MDPs (Brafman & Tennenholtz, 2002)) which results in its worse sample complexity.

With a more complex oracle, Moffle (Modi et al., 2021) is a model-free algorithm for low-rank MDPs, with two additional assumptions: (1) the transition has low *non-negative rank (nnr)*, and (2) reachability in latent states. The first assumption significantly restricts the scope of low-rank MDPs as there are matrices whose nnr is exponentially larger than the rank (Agarwal et al., 2020b). The sample complexity of Moffle can scale $O(d^6|\mathcal{A}|^{13}/(\epsilon^2 \eta^5 (1-\gamma)^5))$, where $\eta$ is the reachability probability, and $1/\eta$ could be as large as $nnr^{1/2}$ (Proposition 4 in Agarwal et al. (2020b)), which essentially means that Moffle has a polynomial dependence on the nnr.

OLIVE (Jiang et al., 2017), Witness rank (Sun et al., 2019) and Bilinear-UCB (Du et al., 2021), when specialized to low-rank MDPs, have slightly tighter dependence on $d$ (e.g., $O(d^2/\epsilon^2)$). But these algorithms are computationally inefficient as they are version space algorithms. Dann et al. (2021) shows that with a policy class, solving a low-rank MDP can take $\Omega(2^d)$ samples. In this work, similar to Witness rank (Sun et al., 2019) and FLAMBE, we use function approximators to model the transition. Thus our positive result is not in contradiction to the result from Dann et al. (2021).

VALOR (Dann et al., 2018), PCID (Du et al., 2019a), HOMER (Misra et al., 2020), RegRL (Foster et al., 2020), and the approach from Feng et al. (2020) are algorithms for block MDPs which is a more restricted setting than low-rank MDPs. These works require additional assumptions such as deterministic transitions (Dann et al., 2018), reachability (Misra et al., 2020; Du et al., 2019a), strong Bellman closure (Foster et al., 2020), and strong unsupervised learning oracles (Feng et al., 2020).

**Offline Setting** We discuss related works in offline RL. Additional related works are discussed in Section A.

Uehara & Sun (2021) obtained similar statistical results for offline RL on low-rank MDPs. Though the sample complexity in their algorithm is slightly tighter, our algorithm is oracle-efficient, while the CPPO algorithm from Uehara & Sun (2021) is a version space algorithm.

Xie et al. (2021) propose a (general) pessimistic model-free algorithm in the offline setting. We can also apply their algorithm to low-rank MDPs and show some finite-sample guarantee. However, it is unclear whether the final bounds in their results can be characterized by the relative condition number only using the true representation, and whether they can compete with history-dependent policies. Thus, our result is still considered superior on low-rank MDPs. The detail is given in Section E.

## 3 PRELIMINARIES

We consider an episodic discounted infinite horizon Markov Decision Process $\mathcal{M} = \langle \mathcal{S}, \mathcal{A}, P, r, \gamma, d_0 \rangle$ specified by a state space $\mathcal{S}$, a discrete action space $\mathcal{A}$, a transition model $P : \mathcal{S} \times \mathcal{A} \to \Delta(\mathcal{S})$, a reward function $r : \mathcal{S} \times \mathcal{A} \to \mathbb{R}$, a discount factor $\gamma \in [0, 1)$, and an initial distribution $d_0 \in \Delta(\mathcal{S})$. To simplify the presentation, we assume $r(s, a)$ and $d_0$ are known (e.g., when $d_0$ is a probability mass only on $s_0$, agent always starts from a fixed initial state $s_0$)[3]. Following prior work (Jiang et al., 2017; Sun et al., 2019), we assume *trajectory reward is normalized*, i.e., for any trajectory $\{s_h, a_h\}_{h=0}^{\infty}$, we have $\sum_{h=0}^{\infty} \gamma^h r(s_h, a_h) \in [0, 1]$. Since the ground truth $P^\star$ is unknown, we need to learn it by interacting with environments in an online manner or utilizing offline data at hand. We remark that the extension of our all results to the finite horizon nonstationary case is straightforward. For example, refer to Zhang et al. (2022).

We use the following notation. Given a policy $\pi : \mathcal{S} \to \Delta(\mathcal{A})$, we define the value function $V_P^\pi(s) = \mathbb{E}\left[\sum_{h=0}^{\infty} \gamma^h r(s_h, a_h) | s_0 = s, P, \pi\right]$ to represent the expected total discounted reward of $\pi$ under $P$ starting at $s$. Similarly, we define the state-action Q function $Q_P^\pi(s, a) := r(s, a) + \gamma \mathbb{E}_{s' \sim P(\cdot|s,a)} V_P^\pi(s')$. The expected total discounted reward of a policy $\pi$ under transition $P$ and reward $r$ is denoted as $V_{P,r}^\pi := \mathbb{E}_{s_0 \sim d_0} V_P^\pi(s_0)$. We define the discounted state-action occupancy distribution $d_P^\pi(s, a) = (1 - \gamma) \sum_{t=0}^{\infty} \gamma^t d_{P,t}^\pi(s, a)$, where $d_{P,t}^\pi(s, a)$ is the probability of $\pi$ visiting $(s, a)$ at time step $t$ under $\pi$ and $P$. We slightly abuse the notation, and denote $d_P^\pi(s)$ as the state visitation, which is equal to $\sum_{a \in \mathcal{A}} d_P^\pi(s, a)$. When $P$ is the true transition model $P^\star$, we drop the subscript and simply use $d^\pi(\cdot)$. Unless otherwise noted, $\Pi$ denotes the class of all polices $\{S \to \Delta(\mathcal{A})\}$. We denote total variation distance of $P_1$ and $P_2$ by $\|P_1 - P_2\|_1$. Finally, given a vector $a$, we define $\|a\|_B = \sqrt{a^\top B a}$. $c_0, c_1, \cdots$ are universal constants.

We study low-rank MDPs defined as follows (Jiang et al., 2017; Agarwal et al., 2020b). The conditions on the upper bounds of the norm of $\phi^\star, \mu^\star$ are just for normalization.

**Definition 1** (Low-rank MDP). *A transition model $P^\star : \mathcal{S} \times \mathcal{A} \to \Delta(\mathcal{A})$ admits a low rank decomposition with rank $d \in \mathbb{N}$ if there exists two embedding functions $\phi^\star \mu^\star$ such that*

$$\forall s, s' \in \mathcal{S}, a \in \mathcal{A} : P^\star(s' \mid s, a) = \mu^\star(s')^\top \phi^\star(s, a)$$

*where $\|\phi^*(s, a)\|_2 \leq 1$ for all $(s, a)$ and for any function $g : \mathcal{S} \to [0, 1]$, $\|\int \mu^\star(s) g(s) \mathrm{d}(s)\|_2 \leq \sqrt{d}$. An MDP is a low rank MDP if $P^\star$ admits such a low rank decomposition.*

Low-rank MDPs capture the latent variable model (Agarwal et al., 2020b) where $\phi^\star(s, a)$ is a distribution over a discrete latent state space $\mathcal{Z}$. The block-MDP model (Du et al., 2019a) is a special instance of the latent variable model with $\phi^\star(s, a)$ being a one-hot encoding vector. Note the linear MDPs (Yang & Wang, 2020; Jin et al., 2020a) assume $\phi^\star$ is known.

Next, we explain two settings: the online learning setting and the offline learning setting. Then, we present our function approximation setup and computational oracles.

**Episodic Online learning** In online learning, our overall goal is to learn a stationary policy $\hat{\pi}$ so that it maximizes $V_{P^\star, r}^{\hat{\pi}}$, where $P^\star$ is the ground truth transition. We assume that we operate under the episodic learning setting where we can only reset to states sampled from the initial distribution $d_0$ (e.g., to emphasize the challenge from exploration, we can consider the special case where we can only reset to a fixed $s_0$). In the episodic setting, given a policy $\pi$, sampling a state $s$ from the

---

[3]Extension to the unknown case is straightforward. Recall the major challenging of RL is due to the unknown transition model.

state visitation $d_P^\pi$ is done by the following *roll-in* procedure: starting at $s_0 \sim d_0$, at every time step $t$, we terminate and return $s_t$ with probability $1 - \gamma$, and otherwise we execute $a_t \sim \pi(s_t)$ and move to $t + 1$, i.e., $s_{t+1} \sim P(\cdot|s_t, a_t)$. Such a sampling procedure is widely used in the policy gradient and policy optimization literature (e.g., (Kakade & Langford, 2002; Agarwal et al., 2021; 2020a)).

**Offline learning**    In the offline RL, we are given a static dataset in the form of quadruples:

$$\mathcal{D} = \{s^{(i)}, a^{(i)}, r^{(i)}, s'^{(i)}\}_{i=1}^n \sim \rho(s, a)\delta(r = r(s, a))P^\star(s' \mid s, a).$$

For simplicity, we assume $\rho = d_{P^\star}^{\pi_b}$, where $\pi_b \in [\mathcal{S} \to \Delta(\mathcal{A})]$ is a fixed behavior policy. We denote $\mathbb{E}_\mathcal{D}[f(s, a, s')] = 1/n \sum_{(s,a,s') \in \mathcal{D}} f(s, a, s')$. To succeed in offline RL, we in general need some coverage property of $\rho$. One common assumption is that $\rho$ globally covers every possible policies' state-action distribution, i.e., $\max_{\pi,s,a} d_{P^\star}^\pi(s, a)/\rho(s, a) < \infty$ (Antos et al., 2008). In this work, we relax such a global coverage assumption and work under the *partial* coverage condition where $\rho$ may not cover distributions of all possible policies. Instead of competing against the optimal policy under the global coverage, we aim to compete against any policies covered by the offline data. In section 5, we precisely define the partial coverage condition using the concept of the relative condition number.

**Function approximation setup and computational oracles**    Since $\mu^\star$ and $\phi^\star$ are unknown, we use function classes to capture them. Our function approximation and computational oracles are *exactly the same as the ones used in* FLAMBE. For completeness, we state the function approximation and computational oracles below.

**Assumption 2.** *We have a model class* $\mathcal{M} = \{(\mu, \phi) : \mu \in \Psi, \phi \in \Phi\}$, *where* $\mu^\star \in \Psi$, $\phi^\star \in \Phi$.

Following the norm bounds on $\mu^\star, \phi^\star$ we similarly assume that the same norm bounds hold for our function approximator, i.e., for any $\mu \in \Psi, \phi \in \Phi$, $\|\phi(s, a)\|_2 \leq 1$, $\forall(s, a)$ and $\|\int \mu(s)g(s)\mathrm{d}(s)\|_2 \leq \sqrt{d}, \forall g : \mathcal{S} \to [0, 1]$, and $\int \mu^\top(s')\phi(s, a)\mathrm{d}(s') = 1$, $\forall(s, a)$.

As for computational oracles, we use a supervised learning style MLE oracle.

**Definition 3** (Maximum Likelihood Oracle (MLE))**.** *Consider the model class* $\mathcal{M}$ *and a dataset* $\mathcal{D}$ *in the form of* $(s, a, s')$*, the MLE oracle returns the maixmum likelihood estimator* $\hat{P} := (\hat{\mu}, \hat{\phi}) = \arg\max_{(\mu,\phi) \in \mathcal{M}} \mathbb{E}_\mathcal{D} \ln(\mu(s')^\top \phi(s, a))$.

We also invoke a planning procedure for *known linear MDPs with potentially nonlinear rewards*, which can be done in polynomial time (we know that online learning in linear MDPs can be done statistically and computationally efficient). Given a reward $r$ and a model $P := (\mu, \phi)$ with $P(s'|s, a) = \mu(s')^\top \phi(s, a)$ (i.e., a *known linear transition* with a *known* feature $\phi$), we can compute the optimal policy $\arg\max_\pi V_{P,r}^\pi$ by standard least square value iteration which uses linear regression. A planning procedure for a known linear MDP is also used in FLAMBE, see Section 5.1 in Agarwal et al. (2020b) how to implement this procedure with polynomial computation complexity.

## 4    REPRESENTATION LEARNING IN ONLINE SETTING

We consider the online episodic learning setting where the agent can only reset based on the initial state distribution $d_0$. To find a near-optimal policy for a low-rank MDP efficiently, we need to carefully interleave representation learning, exploration, and exploitation.

### 4.1    ALGORITHM

We present our algorithm in the online setting in Algorithm 1. We first describe the data collection process. Every iteration, Algorithm 1 rollouts its current policy $\pi$ to collect a tuple $(s, a, s', a', \tilde{s})$ where $s \sim d_{P^\star}^\pi$, $a \sim U(\mathcal{A}), s' \sim P^\star(\cdot|s, a), a' \sim U(\mathcal{A}), \tilde{s} \sim P^\star(\cdot \mid s, a)$ where $U(\mathcal{A})$ is a uniform distribution over actions (note that we take two uniform actions here). Recall that to sample $s \sim d_{P^\star}^\pi$, we start at $s_0 \sim d_0$, at every time step $t$, we terminate and return $s_t$ with probability $1 - \gamma$, and otherwise we execute $a_t \sim \pi(s_t)$ and move to $t + 1$, i.e., $s_{t+1} \sim P^\star(\cdot|s_t, a_t)$. Thus collecting one tuple requires exactly one roll-in, i.e., one trajectory. We can verify that with high probability the roll-in terminates with $\tilde{O}((1 - \gamma)^{-1})$ steps which is often called the effective horizon.

After collecting new data and concatenating it with the existing data, we perform representation learning, i,e, learning a factorization and a representation by MLE (line 6), set the bonus based on

---

**Algorithm 1** UCB-driven representation learning, exploration, and exploitation (REP-UCB)

---

1: **Input:** Regularizer $\lambda_n$, parameter $\alpha_n$, Models $\mathcal{M} = \{(\mu, \phi) : \mu \in \Psi, \phi \in \Phi\}$, Iteration $N$
2: Initialize $\pi_0(\cdot \mid s)$ to be uniform; set $\mathcal{D}_0 = \emptyset, \mathcal{D}'_0 = \emptyset$
3: **for** episode $n = 1, \cdots, N$ **do**
4:     Collect a tuple $(s, a, s', a', \tilde{s})$ with

$$s \sim d_{P^\star}^{\pi_{n-1}}, a \sim U(\mathcal{A}), s' \sim P^\star(\cdot|s,a), a' \sim U(\mathcal{A}), \tilde{s} \sim P^\star(\cdot|s',a')$$

5:     Update datasets by adding triples $(s, a, s')$ and $(s', a', \tilde{s})$:

$$\mathcal{D}_n = \mathcal{D}_{n-1} + \{(s, a, s')\}, \quad \mathcal{D}'_n = \mathcal{D}'_{n-1} + \{(s', a', \tilde{s})\}$$

6:     Learn representation via ERM (i.e., MLE):

$$\hat{P}_n := (\hat{\mu}_n, \hat{\phi}_n) = \arg\max_{(\mu,\phi)\in\mathcal{M}} \mathbb{E}_{\mathcal{D}_n + \mathcal{D}'_n} \left[ \ln \mu^\top(s')\phi(s, a) \right]$$

7:     Update empirical covariance matrix $\hat{\Sigma}_n = \sum_{s,a \in \mathcal{D}_n} \hat{\phi}_n(s, a)\hat{\phi}_n(s, a)^\top + \lambda_n I$
8:     Set the exploration bonus:

$$\hat{b}_n(s, a) := \min\left( \alpha_n \sqrt{\hat{\phi}_n(s, a)^\top \hat{\Sigma}_n^{-1} \hat{\phi}_n(s, a)}, 2 \right) \tag{1}$$

9:     Update policy $\pi_n = \arg\max_\pi V^\pi_{\hat{P}_n, r+\hat{b}_n}$
10: **end for**
11: **Return** $\pi_1, \cdots, \pi_N$

---

the learned feature (Eq. 1), and update the policy via planning inside the learned model with the bonus-enhanced reward (Line 9). Note the learned transition $\hat{P}$ from MLE is linear with respect to the learned feature $\hat{\phi}$, and planning in a known linear MDP is known as computationally efficient (Jin et al., 2020a) (see the explanation after Definition 3 as well).

**Computation of the MLE oracle** The MLE oracle in general could be a non-convex optimization procedure when $\phi$ and $\mu$ are general nonlinear function approximators. However, this is a standard supervised learning ERM oracle and one can easily optimize it via stochastic gradient descent style approaches if $\mu$ and $\phi$ are differentiable. For special cases where the MDP is a tabular MDP, the MLE objective is convex and the optimal solution has closed-form. For linear MDPs (Yang & Wang, 2020) where $P^\star(s'|s, a) = (\psi^\star(s'))^\top M^\star \phi^\star(s, a)$ with known $\mu^\star$ and $\psi^\star$ but unknown $M^\star$, the MLE objective again is convex with respect to parameter $M^\star$. Thus when specializing to specific settings such as tabular MDPs and linear MDPs where computationally efficient approaches exist, Rep-UCB is also provably computationally efficient. In contrast, more general approaches such as Olive (Jiang et al., 2017) are provably computationally inefficient even when specialized to tabular MDPs. We note that our setting does not directly capture the linear MDP setting from Jin et al. (2020a) since there $\mu^\star$ might not be captured by a model class $\Psi$ that has bounded statistical complexity.

### 4.2 ANALYSIS

**Theorem 4** (PAC Bound for REP-UCB). *Fix $\delta \in (0, 1), \epsilon \in (0, 1)$. Let $\hat{\pi}$ be a uniform mixture of $\pi_1, \cdots, \pi_N$ and $\pi^\star := \arg\max_\pi V^\pi_{P^\star, r}$ as the optimal policy. By setting parameters as follows:*

$$\alpha_n = O\left( \sqrt{(|\mathcal{A}| + d^2)\gamma \ln(|\mathcal{M}|n/\delta)} \right), \quad \lambda_n = O\left( d\ln(|\mathcal{M}|n/\delta) \right),$$

*with probability at least $1 - \delta$, we have*

$$V^{\pi^\star}_{P^\star, r} - V^{\hat{\pi}}_{P^\star, r} \leq \epsilon,$$

*where the number of collected samples is at most*

$$O\left( \frac{d^4 |\mathcal{A}|^2 \ln(|\mathcal{M}|/\delta)}{(1-\gamma)^5 \epsilon^2} \cdot \nu \right),$$

*where $\nu$ only contains log terms and the dependence on $|\mathcal{M}|$ is at most $\ln(\ln(|\mathcal{M}|))$.*

The theorem shows that REP-UCB learns in low-rank MDPs in a statistically efficient and oracle-efficient manner. To the best of our knowledge, this algorithm has the best sample complexity among

all oracle efficient algorithms for low-rank MDPs. Extension to continuous function class $\Psi$ and $\Phi$ using statistical complexities such as covering dimension is possible since our analysis *only* uses standard uniform convergence analysis on $\Psi$ and $\Phi$.

**Highlight of the analysis**  Below we highlight our key lemmas and proof techniques.

*First*, why is learning in a low-rank MDP harder than learning in models with linear structures? Unlike standard linear models such as linear MDPs (Yang & Wang, 2020; Jin et al., 2020a), KNRs (Kakade et al., 2020; Abbasi-Yadkori & Szepesvári, 2011; Mania et al., 2020; Song & Sun, 2021), and GP / kernel models (Chowdhury & Gopalan, 2019; Curi et al., 2020), we cannot get uncertainty quantification on the model in a point-wise manner. When models are linear, one can get the following style of point-wise uncertainty quantification for the learned model $\hat{P}$: $\forall s, a : \ell(\hat{P}(\cdot|s,a), P^\star(\cdot|s,a)) \le \sigma(s,a)$ where $\sigma(s,a)$ is the uncertainty measure, and $\ell$ is some distance metric (e.g., $\ell_1$ norm). With proper scaling, the uncertainty measure $\sigma(s,a)$ is then used for the bonus. For example, in linear MDPs (i.e., low-rank MDP with known feature $\phi^\star$), given a dataset $\mathcal{D} = \{s, a, s'\}$, we can learn a non-parametric model $\hat{P}(s'|s,a) := \hat{\mu}(s')^\top \phi^\star(s,a)$ , and get point-wise uncertainty quantification:

$$\forall (s,a), |\int f(s')\hat{\mu}^\top(s')\phi^\star(s,a)\mathrm{d}(s') - \int f(s')\mu^{\star\top}(s')\phi^\star(s,a)\mathrm{d}(s')| \le c\|\phi^\star(s,a)\|_{\Sigma_{\phi^\star}^{-1}} \quad (2)$$

for some family of functions $f : \mathcal{S} \to \mathbb{R}$ with $\Sigma_{\phi^\star} = \sum_{s,a \in \mathcal{D}} \phi^\star(s,a)\phi^\star(s,a) + \lambda I$ (Lykouris et al., 2021; Neu & Pike-Burke, 2020). To set the scaling $c$ properly, since $\phi^\star$ is known a priori, the linear regression analysis applies here, and one can apply Cauchy-Schwarz inequality to the LHS of (2) to pull out $\phi^\star$ and get an upper bound in the form of

$$\underbrace{\|\phi^\star(s,a)\|_{\Sigma_{\phi^\star}^{-1}}}_{(a)} \underbrace{\|\int f(s)\{\hat{\mu}(s) - \mu^\star(s)\}\mathrm{d}(s)\|_{\Sigma_{\phi^\star}}}_{(b)}$$

where $c$ is set to be the linear regression training error measured in the term $(b)$ above.

However, when we jointly learn $\mu$ and $\phi$, since nonlinear function approximation is used, we cannot get point-wise uncertainty quantification via linear regression-based analysis. We stress that our bonus is not designed to capture the uncertainty quantification on the model error between $\hat{P}(\cdot|s,a) = \hat{\mu}^\top\hat{\phi}(s,a)$ and $P^\star(\cdot|s,a) = \mu^{\star\top}\phi^\star(s,a)$ in a point-wise way, which is not tractable as $\hat{P}$ and $P^\star$ does not even share the same representation. Instead, the bonus is carefully designed so that it only provides *near-optimism at the initial state distribution*. This is formalized as follows.

**Lemma 5** (Almost Optimism at the Initial State Distribution). *Set the parameters as in Theorem 4. With probability* $1 - \delta$,

$$\forall n \in [1, \cdots, N], \forall \pi \in \Pi, V^\pi_{\hat{P}_n, r+\hat{b}_n} - V^\pi_{P^\star, r} \ge -c_1 \sqrt{\frac{|\mathcal{A}| \ln(|\mathcal{M}|n/\delta)(1-\gamma)^{-1}}{n}}.$$

We remark that the idea of optimism with respect to the initial state distribution has been used in prior works (Jiang et al., 2017; Sun et al., 2019; Du et al., 2021; Zanette et al., 2020). However, these algorithms are not computationally efficient (i.e., they use version space instead of reward bonus), and their version-space based analysis is different from ours.

**Proof sketch for Lemma 5**  We start by using the simulation lemma (Lemma 20) *inside the learned model* which is important since our bonus $\hat{b}_n$ uses $\hat{\phi}_n$ associated with the learned model $\hat{P}_n$:

$$V^\pi_{\hat{P}_n, r+\hat{b}_n} - V^\pi_{P^\star, r} \ge (1-\gamma)^{-1} \mathbb{E}_{s,a \sim d^\pi_{\hat{P}_n}} \left[ \hat{b}_n(s,a) - \|\hat{P}_n(\cdot|s,a) - P^\star(\cdot|s,a)\|_1 \right],$$

from where we show that $\mathbb{E}_{s,a \sim d^\pi_{\hat{P}_n}} \|\hat{P}(\cdot|s,a) - P^\star(\cdot|s,a)\|_1$ *as a whole* nearly lower bounds the average bonus $\mathbb{E}_{s,a \sim d^\pi_{\hat{P}_n}} [\hat{b}_n(s,a)]$. Thus the proof of optimism is fundamentally different from the proofs in tabular and linear MDPs which are done via induction in a point-wise manner. The detailed procedure is illustrated in Lemma 7 in Appendix B.

*Second*, our bonus is using representation $\hat{\phi}_n$ that is being updated every episode, and our empirical covariance matrix $\hat{\Sigma}_n$ is also updated whenever we update $\hat{\phi}_n$, which means that standard elliptical potential based analysis (i.e., analysis used in linear bandits/MDPs with known features) cannot work here as our feature changes every episode. Instead, in our analysis, we have to keep tracking a potential function that is defined using the unknown ground truth representation $\phi^\star$, i.e., the elliptical potential $\|\phi^\star(s,a)\|_{\Sigma_{\rho_n,\phi^\star}^{-1}}^2$, where

$$\Sigma_{\rho_n,\phi^\star} = n\mathbb{E}_{(s,a)\sim\rho_n}\phi^\star(s,a)\phi^\star(s,a)^\top + \lambda_n I,$$

and $\rho_n(s,a) = \sum_{i=0}^{n-1} d_{P^\star}^{\pi_i}(s,a)/n$. Since this potential function uses the fixed representation $\phi^\star$, we can apply the standard elliptical potential argument to track the progress that our algorithm makes during learning. Below we illustrate the procedure of linking the bonus under $\hat{\phi}_n$ to the potential function $\|\phi^\star(s,a)\|_{\Sigma_{\rho_n,\phi^\star}^{-1}}^2$ defined with respect to the true feature $\phi^\star$.

**Linking bonus under $\hat{\phi}_n$ to the elliptical potential function under $\phi^\star$** With near optimism, using the simulation lemma (Lemma 20) inside the real model, we can upper bound the per-iteration regret as follows:

$$V_{P^\star,r}^{\pi^\star} - V_{P^\star,r}^{\pi_n} \le (1-\gamma)^{-1}\mathbb{E}_{(s,a)\sim d_{P^\star}^{\pi_n}}[\hat{b}_n(s,a) + (1-\gamma)^{-1}f_n(s,a)] + \sqrt{|\mathcal{A}|\zeta_n(1-\gamma)^{-1}},$$

where $\zeta_n = \tilde{O}(1/n)$, and $f_n(s,a) := \|\hat{P}_n(\cdot \mid s,a) - P^\star(\cdot \mid s,a)\|_1$. To connect the first term in the right-hand side of the above inequality to the elliptical potential under the fixed feature $\phi^\star$, we show that for any function $g \in \mathcal{S} \times \mathcal{A} \to [0,B]$ for $B \in \mathbb{R}^+$,

$$\mathbb{E}_{(s,a)\sim d_{P^\star}^{\pi_n}}[g(s,a)] \le (1-\gamma)^{-1}\mathbb{E}_{(s,a)\sim d_{P^\star}^{\pi_n}}\left[\|\phi^\star(s,a)\|_{\Sigma_{\rho_n,\phi^\star}^{-1}}\right]\sqrt{n\gamma|\mathcal{A}|\mathbb{E}_{\rho_n'}[g^2(s,a)] + \gamma\lambda_n dB^2}$$
$$+ \sqrt{(1-\gamma)|\mathcal{A}|\mathbb{E}_{\rho_n'}[g^2(s,a)]},$$

where $\rho_n'(s,a) = 1/n\sum_{i=0}^{n-1} d^{\pi_i}(s)u(a)$ and $u(a) = 1/|\mathcal{A}|$. See Lemma 12 in Appendix B. By substituting $g$ with $\hat{b}_n + f_n/(1-\gamma)$, the first term of the RHS of the above inequality can be upper bounded as:

$$2(1-\gamma)^{-1}\underbrace{\mathbb{E}_{(s,a)\sim d_{P^\star}^{\pi_n}}\left[\|\phi^\star(s,a)\|_{\Sigma_{\rho_n,\phi^\star}^{-1}}\right]}_{(\mathcal{G}_1)}\underbrace{\sqrt{n|\mathcal{A}|\mathbb{E}_{\rho_n'}\left[\frac{f_n^2(s,a)}{(1-\gamma)^2} + \hat{b}_n^2(s,a)\right] + \lambda_n d}}_{(\mathcal{G}_2)}.$$

In the term $(\mathcal{G}_2)$, we expect $n\mathbb{E}_{\rho_n'}[f_n^2(s,a)]$ to be $O(1)$ as $\mathbb{E}_{\rho_n'}[f_n^2(s,a)]$ is in order of $1/n$ due to the fact that it is the generalization bound of the MLE estimator $\hat{P}_n$ which is trained on the data drawn from $\rho_n'$. For $n\mathbb{E}_{\rho_n'}[\hat{b}_n^2(s,a)]$, we expect it to be in the order of $d$ as the (unnormalized) data covariance matrix $\hat{\Sigma}_n$ in the bonus $\hat{b}_n$ uses training data from $\rho_n'$, i.e., we are measuring the expected bonus under the training distribution. In other words, the term $(\mathcal{G}_2)$ scales in order of poly$(d)$. For the term $(\mathcal{G}_1)$, since it contains the potential function based on $\phi^\star$, the sum of the term $(\mathcal{G}_1)$ over all episodes can be controlled by the standard elliptical potential argument (see Lemma 18 and Lemma 19). This concludes the proof sketch of our main theorem.

**In summary**, our analysis relies on the standard idea of optimism in the face of uncertainty, but with novel techniques to achieve optimism under nonlinear function approximation with the MLE supervised learning style generalization bound, and to track regret under changing representations.

## 5 REPRESENTATION LEARNING IN OFFLINE SETTING

In this section, we study representation learning in the offline setting. We consider the setting where the offline data does not have a full global coverage. We present our algorithm *Lower Confidence Bound driven Representation Learning in offline RL* (REP-LCB) in Algorithm 2. Our proposed algorithm consists of three parts. The first part is MLE which learns a model $\hat{P}$ and a representation $\hat{\phi}$. The second part is the construction of a penalty term $\hat{b}$. Using the learned representation $\hat{\phi}$, we use a standard bonus in linear bandits as the penalty term as if $\hat{\phi}$ were the true feature. The third part is planning with the learned model $\hat{P}$ and reward $r - \hat{b}$.

---

**Algorithm 2** LCB-driven Representation Learning in offline RL (REP-LCB)

---

1: **Input:** Regularizer $\lambda$, Parameter $\alpha$, Model classes $\mathcal{M} = \{\mu^\top \phi : \mu \in \Psi, \phi \in \Phi\}$, Dataset $\mathcal{D}$.
2: Learn a model $\hat{P}$ by MLE: $\hat{P} = \hat{\mu}^\top \hat{\phi} = \arg\max_{P \in \mathcal{M}} \mathbb{E}_{\mathcal{D}}[\ln P(s' \mid s, a)]$.
3: Set the empirical covariance matrix $\hat{\Sigma} = \sum_{(s,a) \in \mathcal{D}} \hat{\phi}(s,a)\hat{\phi}^\top(s,a) + \lambda I$.
4: Set the reward penalty:
$$\hat{b}(s,a) = \min\left(\alpha\sqrt{\hat{\phi}(s,a)^\top \hat{\Sigma}^{-1} \hat{\phi}(s,a)}, 2\right).$$
5: Solve $\hat{\pi} = \arg\max_\pi V^\pi_{\hat{P}, r - \hat{b}}$.

---

We present the PAC guarantee of REP-LCB. Before proceeding, we define a relative condition number as a mean to measure the deviation between a comparator policy $\pi$ and the offline data:

$$C^\star_\pi = \sup_{x \in \mathbb{R}^d} \frac{x^\top \mathbb{E}_{d^\pi_{P^\star}}[\phi^\star(s,a)\phi^{\star\top}(s,a)]x}{x^\top \mathbb{E}_\rho[\phi^\star(s,a)\phi^{\star\top}(s,a)]x}.$$

In the special case where the MDP is just a tabular MDP (i.e., $\phi^\star$ is a one-hot encoding vector), this is reduced to a density ratio $C^\star_\infty = \max_{s,a} d^\pi_{P^\star}(s,a)/\rho(s,a)$. The relative condition number $C^\star_\pi$ is always no larger than the density ratio and could be much smaller for MDPs with large state spaces. Note that we quantify the relative condition number using the unknown true representation $\phi^\star$. With the above setup, now we are ready to state the main theorem for REP-LCB.

**Theorem 6** (PAC Bound for REP-LCB). *Let $\omega = \max_{a,s}(1/\pi_b(a \mid s))$. Denote $\hat{\pi}$ as the output of* REP-LCB. *There exists a set of parameters such that with probability at least $1 - \delta$, for* any *policy $\pi$ (including history-dependent non-Markovian policies),*

$$V^\pi_{P^\star, r} - V^{\hat{\pi}}_{P^\star, r} \leq c\sqrt{\frac{d^4 \omega^2 C^\star_\pi \log(|\mathcal{M}|/\delta)}{(1-\gamma)^4 n}}.$$

See Theorem 14 in Appendix B for the detailed parameters. We explain several implications. First of all, this theorem shows that we can uniformly compete with any policy including history-dependent non-Markovian policies [4] satisfying the partial coverage $C^\star_\pi < \infty$. Particularly, if the optimal policy $\pi^\star$ is covered by the offline data, i.e., $C^\star_{\pi^\star} < \infty$, then our algorithm is able to compete against it [5]. Note that assuming offline data covers $\pi^\star$ is still a weaker assumption than the global coverage such as $\sup_\pi \sup_{(s,a)} d^\pi_{P^\star}(s,a)/\rho(s,a)$ in prior offline RL works (Antos et al., 2008; Chen & Jiang, 2019). Second, our coverage condition is measured by a relative condition number defined using the unknown ground truth representation $\phi^\star$ but not depending on other features. Prior works that use relative condition numbers as measures of coverage are restricted to the settings where the ground truth representation $\phi^\star$ is known (Jin et al., 2020b; Chang et al., 2021; Zanette et al., 2021b).

To sum up, our algorithm is the first oracle efficient algorithm which does not need to know $\phi^\star$, and requires partial coverage only in terms of $\phi^\star$. Note while Uehara & Sun (2021) has a similar guarantee on low-rank MDPs, their algorithm is not oracle-efficient as it is a version space algorithm.

## 6 CONCLUSION

We study online/offline RL on low-rank MDPs, where the ground truth feature is not known a priori. For online RL, our new algorithm REP-UCB significantly improves the sample complexity of the piror state-of-the-art algorithm FLAMBE in all parameters while using the same computational oracles. REP-UCB has the best sample complexity among existing oracle efficient algorithms for low-rank MDPs by a margin. Comparing to prior representation learning works on low-rank MDPs and block MDPs that rely on a forward step-by-step reward-free exploration framework, our algorithm interleaves representation learning, exploration, and exploitation together, and learns a single stationary policy. For offline RL, our new algorithm REP-LCB is the first oracle efficient algorithm for low-rank MDPs that has a PAC guarantee under a partial coverage condition measured by the relative condition number defined with the true feature representation.

---

[4] Given $\pi = \{\pi_i\}_{i=0}^\infty$ where $\pi_i$ depends on $s_0, a_0, \dots s_i$, $V^\pi_{P^\star, r}$ and $d^\pi_{P^\star}(s,a)$ are still well-defined.
[5] We also require $\omega < \infty$, which is a mild assumption since it does not involve $P^\star$. Indeed, it is much weaker than the global coverage type assumption $1/\rho(s,a) < \infty, \forall(s,a)$.

ACKNOWLEDGMENTS

The authors would like to thank Alekh Agarwal, Praneeth Netrapalli and Ming Yin for valuable feedback.

Masatoshi Uehara is partially supported by Masason foundation.

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

## A    MORE RELATED WORK

Here, we mention several additional related works.

**Online setting**    We mention works that tackle representation learning in quite different settings.

Hao et al. (2021b) consider feature selection in sparse linear MDPs with a given exploratory distribution. Zhang et al. (2021a) consider how to choose the best representation among *correct representations* inspired by Papini et al. (2021) (i.e., the MDP is a linear MDP under any representation in the function class $\Phi$). Thus, it still falls into the linear function approximation setting. In contrast, we only assume the MDP is linear under *some* unknown $\phi^\star \in \Phi$.

**Offline setting**    In addition to the two work we mentioned, the pessimistic approach in offline RL has been extensively investigated. Empirically, it can work on simulation control tasks (Kidambi et al., 2020; Yu et al., 2020; Kumar et al., 2020; Liu et al., 2020; Chang et al., 2021). On the theoretical side, pessimism allows us to obtain the PAC guarantee on various models when a comparator policy is covered by offline data in some forms (Jin et al., 2020b; Rashidinejad et al., 2021; Yin et al., 2021; Zanette et al., 2021b; Zhang et al., 2021b; Chang et al., 2021). However, these algorithms and their analysis rely on a *known representation* and linear function approximation.

We mention works that tackle representation learning from different viewpoints and settings. Lu et al. (2021) consider multitask representation learning under a generative model assumption. Hao et al. (2021a) study the feature selection problem in sparse linear MDPs and Ni et al. (2021) study dimensionality reduction in a given kernel space, under the assumption that the offline data admits some form of full coverage condition. Shah et al. (2020) studies learning on the assumption that the optimal Q-function admits a low-rank structure on the generative model setting.

## B    PROOF OF THE THEORETICAL PROPERTY OF REP-UCB

**Notation**    We summarize the notations we frequently use. First of all, hereafter, we assume $c_0, c_1, \cdots$, are some universal constants, and the notation

$$f(1/(1-\gamma), |\mathcal{A}|, \ln(1/\delta), \ln(|\mathcal{M}|), d, n) \lesssim g(1/(1-\gamma), |\mathcal{A}|, \ln(1/\delta), \ln(|\mathcal{M}|), d, n)$$

means there exists some constant $c_1 > 0$, such that

$$f(1/(1-\gamma), |\mathcal{A}|, \ln(1/\delta), \ln(|\mathcal{M}|), d, n) \leq c_1 g(1/(1-\gamma), |\mathcal{A}|, \ln(1/\delta), \ln(|\mathcal{M}|), d, n)$$

for any $0 \leq \gamma < 1, |\mathcal{A}|, \ln(1/\delta), \ln(|\mathcal{M}|), d, n$.

We define

$$\rho_n(s) := \frac{1}{n} \sum_{i=0}^{n-1} d_{P^\star}^{\pi_i}(s).$$

With slight abuse of notation, we overload the above notation and use $\rho_n$ for $1/n \sum_{i=0}^{n-1} d_{P^\star}^{\pi_i}(s, a)$. Next, define $\rho_n' \in [\mathcal{S} \to \mathbb{R}]$ as a marginal distribution of $s'$ for a triple

$$(s, a, s') \sim \rho_n(s) U(a) P^\star(s' \mid s, a).$$

We define three matrices as follows:

$$\Sigma_{\rho_n \times U(\mathcal{A}), \phi} = n \mathbb{E}_{s \sim \rho_n, a \sim U(\mathcal{A})}[\phi(s, a)\phi^\top(s, a)] + \lambda_n I,$$

$$\Sigma_{\rho_n, \phi} = n \mathbb{E}_{(s,a) \sim \rho_n}[\phi(s, a)\phi^\top(s, a)] + \lambda_n I,$$

$$\hat{\Sigma}_{n, \phi} = n \mathbb{E}_{(s,a) \sim \mathcal{D}_n}[\phi\phi^\top] + \lambda_n I.$$

Note that for a fixed $\phi$, $\hat{\Sigma}_{n, \phi}$ is an unbiased estimate of $\Sigma_{\rho_n \times U(\mathcal{A}), \phi}$.

**Optimism** First, we prove the optimism at the initial distribution. This is proved by using a simulation lemma inside the learned model which is important since both the bonus and the learned model use $\hat\phi$. In high level, we will show that the expected bonus $\mathbb{E}_{s,a\sim d_{\hat{P}_n}^\pi}\, \hat{b}_n(s,a)$ is in the same order of the expected model error $\mathbb{E}_{s,a\sim d_{\hat{P}_n}^\pi}\|\hat{P}_n(\cdot|s,a) - P^\star(\cdot|s,a)\|_1$. Note that the expectation is with respect to $d_{\hat{P}_n}^\pi$.

**Lemma 7** (Almost Optimism at the Initial Distribution). *Consider an episode $n\,(1 \le n \le N)$ and set*

$$\alpha_n = O(\sqrt{(|\mathcal{A}| + d^2)\,\gamma\ln(|\mathcal{M}|n/\delta)}), \quad \lambda_n = O\left(d\ln(|\mathcal{M}|n/\delta)\right), \zeta_n = O\left(\frac{\ln(|\mathcal{M}|n/\delta)}{n}\right).$$

*With probability $1 - \delta$, we have*

$$\forall n \in [1, \cdots, N], \forall \pi \in \Pi, V_{\hat{P}_n, r + \hat{b}_n}^\pi - V_{P^\star, r}^\pi \ge -\sqrt{(1-\gamma)^{-1}|\mathcal{A}|\zeta_n}.$$

*Proof.* In this proof, letting $f_n(s,a) = \|\hat{P}_n(\cdot \mid s,a) - P^\star(\cdot \mid s,a)\|_1$, we condition on the event

$$\forall n, \quad \mathbb{E}_{s\sim\rho_n, a\sim U(\mathcal{A})}[f_n^2(s,a)] \le \zeta_n, \quad \mathbb{E}_{s\sim\rho_n', a\sim U(\mathcal{A})}[f_n^2(s,a)] \le \zeta_n,$$

$$\forall n, \forall \phi, \|\phi(s,a)\|_{\hat\Sigma_n^{-1}, \phi} = \Theta(\|\phi(s,a)\|_{\Sigma_{\rho_n \times U(\mathcal{A}), \phi}^{-1}}).$$

From Lemma 10 and Lemma 17, this event happens with probability $1 - \delta$. Then, for any policy $\pi$, from simulation lemma 20,

$$(1-\gamma)(V_{\hat{P}_n, r+\hat{b}_n}^\pi - V_{P^\star, r}^\pi)$$
$$= \mathbb{E}_{(s,a)\sim d_{\hat{P}_n}^\pi}\left[\hat{b}_n(s,a) + \gamma\mathbb{E}_{s'\sim\hat{P}_n(s,a)}\left[V_{P^\star, r}^\pi(s')\right] - \gamma\mathbb{E}_{s'\sim P^\star(s,a)}\left[V_{P^\star, r}^\pi(s')\right]\right]$$
$$\gtrsim \mathbb{E}_{(s,a)\sim d_{\hat{P}_n}^\pi}\left[\min\left(\alpha_n\|\hat\phi_n(s,a)\|_{\Sigma_{\rho_n \times U(\mathcal{A}), \hat\phi_n}^{-1}}, 2\right) + \gamma\mathbb{E}_{s'\sim\hat{P}_n(s,a)}\left[V_{P^\star, r}^\pi(s')\right] - \gamma\mathbb{E}_{s'\sim P^\star(s,a)}\left[V_{P^\star, r}^\pi(s')\right]\right]$$
$$\tag{3}$$

where in the last step, we replaced the empirical covariance by the population covariance. Note the notation $\lesssim$ is up to universal constants. Here, since $\|V_{P^\star, r}^\pi\|_\infty \le 1$ (since we assume trajectory-wise total reward is normalized between $[0, 1]$), we have:

$$\left|\mathbb{E}_{(s,a)\sim d_{\hat{P}_n}^\pi}\left\{\mathbb{E}_{s'\sim\hat{P}_n(s,a)}\left[V_{P^\star, r}^\pi(s')\right] - \mathbb{E}_{s'\sim P^\star(s,a)}\left[V_{P^\star, r}^\pi(s')\right]\right\}\right| \le \mathbb{E}_{(s,a)\sim d_{\hat{P}_n}^\pi}\left\{f_n(s,a)\right\}.$$

The above is further bounded by Lemma 11:

$$|\mathbb{E}_{(s,a)\sim d_{\hat{P}_n}^\pi}\left\{f_n(s,a)\right\}| \le \mathbb{E}_{(\tilde{s},\tilde{a})\sim d_{\hat{P}_n}^\pi}\|\hat\phi_n(\tilde{s},\tilde{a})\|_{\Sigma_{\rho_n \times U(\mathcal{A}), \hat\phi_n}^{-1}}\sqrt{\gamma}\sqrt{\left\{n|\mathcal{A}|\mathbb{E}_{s\sim\rho_n', a\sim U(\mathcal{A})}\left[f_n^2(s,a)\right]\right\} + 4\lambda_n d + 4n\zeta_n}$$
$$+ \sqrt{(1-\gamma)|\mathcal{A}|\mathbb{E}_{s\sim\rho_n, a\sim U(\mathcal{A})}\left[f_n^2(s,a)\right]}.$$

Then,

$$\mathbb{E}_{(s,a)\sim d_{\hat{P}_n}^\pi}\left\{f_n(s,a)\right\} \lesssim \sqrt{\alpha_n'}\mathbb{E}_{(\tilde{s},\tilde{a})\sim d_{\hat{P}_n}^\pi}\|\hat\phi_n(\tilde{s},\tilde{a})\|_{\Sigma_{\rho_n \times U(\mathcal{A}), \hat\phi_n}^{-1}} + \sqrt{|\mathcal{A}|\zeta_n(1-\gamma)}. \tag{4}$$

where

$$\alpha_n' = \gamma\{n|\mathcal{A}|\zeta_n + \lambda_n d + n\zeta_n\} \lesssim \gamma\left(|\mathcal{A}| + d^2\right)\ln(|\mathcal{M}|n/\delta).$$

Note we here use $f_n(s,a) \le 2, \mathbb{E}_{s\sim\rho_n, a\sim U(\mathcal{A})}[f_n(s,a)^2] \le \zeta_n$ and $\mathbb{E}_{s\sim\rho_n', a\sim U(\mathcal{A})}[f_n(s,a)^2] \le \zeta_n$.

Combining all things together,

$$\left|\mathbb{E}_{(s,a)\sim d_{\hat{P}_n}^\pi}\left\{\mathbb{E}_{s'\sim\hat{P}_n(s,a)}\left[V_{P^\star, r}^\pi(s')\right] - \mathbb{E}_{s'\sim P^\star(s,a)}\left[V_{P^\star, r}^\pi(s')\right]\right\}\right| \le 2\mathbb{E}_{(s,a)\sim d_{\hat{P}_n}^\pi}\left\{f_n(s,a)\right\}$$
$$\lesssim \sqrt{\alpha_n'}\mathbb{E}_{(\tilde{s},\tilde{a})\sim d_{\hat{P}_n}^\pi}\|\hat\phi_n(\tilde{s},\tilde{a})\|_{\Sigma_{\rho_n \times U(\mathcal{A}), \hat\phi_n}^{-1}} + \sqrt{(1-\gamma)|\mathcal{A}|\zeta_n}$$
$$\le \alpha_n\mathbb{E}_{(\tilde{s},\tilde{a})\sim d_{\hat{P}_n}^\pi}\|\hat\phi_n(\tilde{s},\tilde{a})\|_{\Sigma_{\rho_n \times U(\mathcal{A}), \hat\phi_n}^{-1}} + \sqrt{(1-\gamma)|\mathcal{A}|\zeta_n}, \quad \text{where } \alpha_n := \sqrt{\alpha_n'}. \tag{5}$$

Going back to (3), we have

$$(1-\gamma)(V^{\pi}_{\hat{P}_n,r+\hat{b}_n} - V^{\pi}_{P^\star,r})$$

$$\gtrsim \mathbb{E}_{(s,a)\sim d^{\pi}_{\hat{P}_n}}\left[\min\left(\alpha_n\|\hat{\phi}_n(s,a)\|_{\Sigma^{-1}_{\rho_n\times U(\mathcal{A}),\hat{\phi}_n}}, 2\right) + \gamma\mathbb{E}_{s'\sim\hat{P}_n(s,a)}\left[V^{\pi}_{P^\star,r}(s')\right] - \gamma\mathbb{E}_{s'\sim P^\star(s,a)}\left[V^{\pi}_{P^\star,r}(s')\right]\right]$$

$$\geq \mathbb{E}_{(s,a)\sim d^{\pi}_{\hat{P}_n}}\left[\min\left(\alpha_n\|\hat{\phi}_n(s,a)\|_{\Sigma^{-1}_{\rho_n\times U(\mathcal{A}),\hat{\phi}_n}}, 2\right) - \min\left(\alpha_n\|\hat{\phi}_n(s,a)\|_{\Sigma^{-1}_{\rho_n\times U(\mathcal{A}),\hat{\phi}_n}} + \sqrt{(1-\gamma)|\mathcal{A}|\zeta_n}, 2\right)\right]$$

$$\geq -\sqrt{(1-\gamma)|\mathcal{A}|\zeta_n}.$$

From the second line to the third line, we again use $\|V^{\pi}_{P^\star,r}\|_\infty = O(1)$ and (4). This concludes the proof. □

Next, we obtain the upper bound of $\sum_{n=0}^{N} V^{\pi^\star}_{P^\star,r} - V^{\pi_n}_{P^\star,r}$. Recall $\pi^\star$ is the optimal policy. Though this form is the same as a standard regret form, since we are not exactly deploying $\pi_n$ in episode $n$ (recall that we play a uniform action at the end of the episode), we cannot get the regret guarantee. However, it suffices for the PAC guarantee.

**Lemma 8** (Regret). *With probability $1-\delta$, we have*

$$\sum_{n=1}^{N} V^{\pi^\star}_{P^\star,r} - V^{\pi_n}_{P^\star,r} \lesssim \sqrt{N\ln\left(1+\frac{N}{\lambda_1}\right)\ln(N|\mathcal{M}|/\delta)}\frac{|\mathcal{A}|d^2}{(1-\gamma)}.$$

*Proof.* Similar to Lemma 7, letting $f_n(s,a) = \|\hat{P}_n(\cdot\mid s,a) - P^\star(\cdot\mid s,a)\|_1$, we condition on the event

$$\forall n, \quad \mathbb{E}_{s\sim\rho_n,a\sim U(\mathcal{A})}[f_n^2(s,a)] \leq \zeta_n, \quad \forall\phi, \|\phi(s,a)\|_{\hat{\Sigma}_n^{-1},\phi} = \Theta(\|\phi(s,a)\|_{\Sigma^{-1}_{\rho_n\times\mathcal{U},\phi}}). \tag{6}$$

From Lemma 10 and Lemma 17, this event happens with probability $1-\delta$.

For any fixed episode $n$ and any policy $\pi$, we have

$$V^{\pi^\star}_{P^\star,r} - V^{\pi_n}_{P^\star,r}$$

$$\leq V^{\pi^\star}_{\hat{P}_n,r+\hat{b}_n} - V^{\pi_n}_{P^\star,r} + \sqrt{|\mathcal{A}|\zeta_n(1-\gamma)^{-1}} \tag{Lemma 7}$$

$$\leq V^{\pi_n}_{\hat{P}_n,r+\hat{b}_n} - V^{\pi_n}_{P^\star,r} + \sqrt{|\mathcal{A}|\zeta_n(1-\gamma)^{-1}} \tag{$\pi_n = \arg\max_\pi V^{\pi}_{\hat{P}_n,r+\hat{b}_n}$}$$

$$= (1-\gamma)^{-1}\mathbb{E}_{(s,a)\sim d^{\pi_n}_{P^\star}}[\hat{b}_n(s,a) + \gamma\mathbb{E}_{\hat{P}_n(s'|s,a)}[V^{\pi_n}_{\hat{P}_n,r+\hat{b}_n}(s')] - \gamma\mathbb{E}_{P^\star(s'|s,a)}[V^{\pi_n}_{\hat{P}_n,r+\hat{b}_n}(s')]] + \sqrt{|\mathcal{A}|\zeta_n(1-\gamma)^{-1}}.$$

We use the 2nd form of simulation Lemma 20 in the last display.

Then, noting $\|\hat{b}_n\|_\infty \leq 2$, we have $\|V^{\pi_n}_{\hat{P}_n,r+\hat{b}_n}\|_\infty \leq 2/(1-\gamma)$. Combining this fact with the above expansion, we have

$$(V^{\pi^\star}_{P^\star,r} - V^{\pi_n}_{P^\star,r})$$

$$\leq (1-\gamma)^{-1}\underbrace{\mathbb{E}_{(s,a)\sim d^{\pi_n}_{P^\star}}[\hat{b}_n(s,a)]}_{(a)} + \left(\frac{2}{(1-\gamma)^2}\right)\underbrace{\mathbb{E}_{(s,a)\sim d^{\pi_n}_{P^\star}}[f_n(s,a)]}_{(b)} + \sqrt{|\mathcal{A}|\zeta_n(1-\gamma)^{-1}}. \tag{7}$$

First, we calculate the first term (a) in Inequality 7. Following Lemma 12 and noting the bonus $\hat{b}_n$ is $O(1)$, we have

$$\mathbb{E}_{(s,a)\sim d^{\pi_n}_{P^\star}}\left[\hat{b}_n(s,a)\right]$$

$$\lesssim \mathbb{E}_{(s,a)\sim d^{\pi_n}_{P^\star}}\left[\min\left(\alpha_n\|\hat{\phi}_n(s,a)\|_{\Sigma^{-1}_{\rho_n\times\mathcal{U}(\mathcal{A}),\hat{\phi}_n}}, 2\right)\right] \tag{From (6)}$$

$$\lesssim \mathbb{E}_{(\tilde{s},\tilde{a})\sim d^{\pi_n}_{P^\star}}\|\phi^\star(\tilde{s},\tilde{a})\|_{\Sigma^{-1}_{\rho_n,\phi^\star}}\sqrt{n\gamma|\mathcal{A}|\alpha_n^2\mathbb{E}_{s\sim\rho_n,a\sim U(\mathcal{A})}\left[\|\hat{\phi}_n(s,a)\|^2_{\Sigma^{-1}_{\rho_n\times U(\mathcal{A}),\hat{\phi}_n}}\right]} + d\gamma\lambda_n$$

$$+ \sqrt{|\mathcal{A}|\alpha_n^2\mathbb{E}_{s\sim\rho_n,a\sim U(\mathcal{A})}\left[\|\hat{\phi}_n(s,a)\|^2_{\Sigma^{-1}_{\rho_n\times U(\mathcal{A}),\hat{\phi}_n}}\right]}(1-\gamma).$$

Note that we use the fact that $B = 2$ when applying Lemma 12. In addition, we have

$$n\mathbb{E}_{s\sim\rho_n,a\sim U(\mathcal{A})}\left[\|\hat{\phi}_n(s,a)\|^2_{\Sigma^{-1}_{\rho_n\times U(\mathcal{A}),\hat{\phi}_n}}\right] = n\operatorname{Tr}(\mathbb{E}_{\rho_n\times U(\mathcal{A})}[\hat{\phi}_n\hat{\phi}_n^\top]\{n\mathbb{E}_{\rho_n\times U(\mathcal{A})}[\hat{\phi}_n\hat{\phi}_n^\top] + \lambda_n I\}^{-1}) \le d.$$

Then,

$$\mathbb{E}_{(s,a)\sim d^{\pi_n}_{P^\star}}\left[\hat{b}_n(s,a)\right] \le \mathbb{E}_{(\tilde{s},\tilde{a})\sim d^{\pi_n}_{P^\star}}\|\phi^\star(\tilde{s},\tilde{a})\|_{\Sigma^{-1}_{\rho_n,\phi^\star}}\sqrt{\gamma d|\mathcal{A}|\alpha_n^2 + \gamma d\lambda_n} + \sqrt{d|\mathcal{A}|\alpha_n^2(1-\gamma)/n}.$$

Second, we calculate the term (b) in inequality 7. Following Lemma 12 and noting $f_n^2(s,a)$ is upper-bounded by 4 (i.e., $B = 4$ in Lemma 12), we have

$$\mathbb{E}_{(s,a)\sim d^{\pi_n}_{P^\star}}[f_n(s,a)]$$
$$\le \mathbb{E}_{(\tilde{s},\tilde{a})\sim d^{\pi_n}_{P^\star}}\|\phi^\star(\tilde{s},\tilde{a})\|_{\Sigma^{-1}_{\rho_n,\phi^\star}}\sqrt{\{n|\mathcal{A}|\gamma\mathbb{E}_{s\sim\rho_n,a\sim U(\mathcal{A})}[f_n^2(s,a)]\} + 4\gamma\lambda_n d}$$
$$+ \sqrt{|\mathcal{A}|\mathbb{E}_{s\sim\rho_n,a\sim U(\mathcal{A})}[f_n^2(s,a)(1-\gamma)]}$$
$$\le \mathbb{E}_{(\tilde{s},\tilde{a})\sim d^{\pi_n}_{P^\star}}\|\phi^\star(\tilde{s},\tilde{a})\|_{\Sigma^{-1}_{\rho_n,\phi^\star}}\sqrt{n|\mathcal{A}|\gamma\zeta_n + 4\gamma\lambda_n d} + \sqrt{|\mathcal{A}|\zeta_n(1-\gamma)}$$
$$\le \mathbb{E}_{(\tilde{s},\tilde{a})\sim d^{\pi_n}_{P^\star}}\|\phi^\star(\tilde{s},\tilde{a})\|_{\Sigma^{-1}_{\rho_n,\phi^\star}}\alpha_n + \sqrt{|\mathcal{A}|\zeta_n(1-\gamma)},$$

where in the second inequality, we use $\mathbb{E}_{s\sim\rho_n,a\sim U(\mathcal{A})}[f_n^2(s,a)] \le \zeta_n$, and in the last line, recall $\sqrt{\gamma}\sqrt{n|\mathcal{A}|\zeta_n + \lambda_n d + n\zeta_n} \lesssim \alpha_n$.

Then, by combining the above calculation of the term (a) and term (b) in inequality 7, we have:

$$V^{\pi^\star}_{P^\star,r} - V^{\pi_n}_{P^\star,r} \lesssim \frac{1}{(1-\gamma)}\left(\mathbb{E}_{(\tilde{s},\tilde{a})\sim d^{\pi_n}_{P^\star}}\|\phi^\star(\tilde{s},\tilde{a})\|_{\Sigma^{-1}_{\rho_n,\phi^\star}}\sqrt{d|\mathcal{A}|\alpha_n^2 + d\lambda_n} + \sqrt{\frac{d|\mathcal{A}|\alpha_n^2(1-\gamma)}{n}}\right)$$
$$+ \frac{1}{(1-\gamma)^2}\left(\mathbb{E}_{(\tilde{s},\tilde{a})\sim d^{\pi_n}_{P^\star}}\|\phi^\star(\tilde{s},\tilde{a})\|_{\Sigma^{-1}_{\rho_n,\phi^\star}}\alpha_n + \sqrt{|\mathcal{A}|\zeta_n(1-\gamma)}\right).$$

Hereafter, we take the dominating term out. First, recall

$$\alpha_n \lesssim \sqrt{\{|\mathcal{A}| + d^2\}\ln(N|\mathcal{M}|/\delta))} \lesssim \sqrt{|\mathcal{A}|d^2\ln(N|\mathcal{M}|/\delta)}.$$

Second, we also use

$$\sum_{n=1}^N \mathbb{E}_{(\tilde{s},\tilde{a})\sim d^{\pi_n}_{P^\star}}\|\phi^\star(\tilde{s},\tilde{a})\|_{\Sigma^{-1}_{\rho_n,\phi^\star}} \le \sqrt{N\sum_{n=1}^N \mathbb{E}_{(\tilde{s},\tilde{a})\sim d^{\pi_n}_{P^\star}}[\phi^\star(\tilde{s},\tilde{a})^\top\Sigma^{-1}_{\rho_n,\phi^\star}\phi^\star(\tilde{s},\tilde{a})]}$$
$$\text{(CS inequality)}$$
$$\lesssim \sqrt{N\left(\ln\det(\sum_{n=1}^N \mathbb{E}_{(\tilde{s},\tilde{a})\sim d^{\pi_n}_{P^\star}}[\phi^\star(\tilde{s},\tilde{a})\phi^\star(\tilde{s},\tilde{a})^\top]) - \ln\det(\lambda_1 I)\right)}$$
$$\text{(Lemma 18 and } \lambda_1 \le \cdots \le \lambda_N)$$
$$\le \sqrt{dN\ln\left(1 + \frac{N}{d\lambda_1}\right)}.$$
$$\text{(Potential function bound, Lemma 19 noting } \|\phi^\star(s,a)\|_2 \le 1 \text{ for any } (s,a).)$$

Finally,

$$
\begin{aligned}
\sum_{n=1}^{N} V_{P^\star,r}^{\pi} - V_{P^\star,r}^{\pi_n} &\lesssim \frac{1}{(1-\gamma)} \left( \sqrt{dN \ln\left(1 + \frac{N}{d\lambda_1}\right)} \sqrt{d|\mathcal{A}|\alpha_N^2 + d\lambda_N} + \sum_{n=1}^{N} \sqrt{\frac{d|\mathcal{A}|\alpha_n^2(1-\gamma)}{n}} \right) \\
&\quad + \frac{1}{(1-\gamma)^2} \left( \sqrt{dN \ln\left(1 + \frac{N}{d\lambda_1}\right)} \alpha_N + \sum_{n=1}^{N} \sqrt{|\mathcal{A}|\zeta_n(1-\gamma)} \right) \\
&\lesssim \frac{1}{(1-\gamma)} \sqrt{dN \ln\left(1 + \frac{N}{d\lambda_1}\right)} \sqrt{d|\mathcal{A}|\alpha_N^2} + \frac{1}{(1-\gamma)^2} \sqrt{dN \ln\left(1 + \frac{N}{d\lambda_1}\right)} \alpha_N \\
&\qquad\qquad\qquad \text{(Some algebra. We take the dominating term out.)} \\
&\lesssim \sqrt{dN \ln\left(1 + \frac{N}{d\lambda_1}\right)} \frac{|\mathcal{A}|d^{3/2} \ln^{1/2}(N|\mathcal{M}|/\delta)}{(1-\gamma)^2}.
\end{aligned}
$$

This concludes the proof. $\qquad\square$

Using Lemma 8, we can immediately obtain the PAC guarantee.

**Theorem 9** (PAC guarantee of REP-UCB). *By interacting with the environment for a number of steps at most*

$$
N \log(N/\delta), \quad N := O\left( \frac{d^4|\mathcal{A}|^2 \ln(|\mathcal{M}|/\delta)}{(1-\gamma)^5\epsilon^2} \ln^2\left(1 + \frac{d^4|\mathcal{A}|^2 \ln(|\mathcal{M}|/\delta)}{(1-\gamma)^5\epsilon^2}\right) \right).
$$

*with probability $1 - \delta$, we can ensure $V_{P^\star,r}^{\pi^\star} - V_{P^\star,r}^{\hat{\pi}} \leq \epsilon$.*

*Proof.* From Lemma 8 and Lemma 22, when $N$ is

$$
O\left( \frac{d^4|\mathcal{A}|^2 \ln(|\mathcal{M}|/\delta)}{(1-\gamma)^4\epsilon^2} \ln^2\left(1 + \frac{d^4|\mathcal{A}|^2 \ln(|\mathcal{M}|/\delta)}{(1-\gamma)^4\epsilon^2}\right) \right),
$$

with probability $1 - \delta$, we can ensure

$$
\frac{1}{N} \sum_{n=1}^{N} V_{P^\star,r}^{\pi^\star} - V_{P^\star,r}^{\pi_n} \leq \epsilon.
$$

With probability $1-\delta$, we need $(1-\gamma)^{-1} \ln(1/\delta)$ interactions with the environment to get one tuple $(s, a, s', a', \tilde{s})$ from one roll-in of $\pi$. Thus, the total sample complexity is $O(N(1-\gamma)^{-1} \ln(N/\delta))$.

$\qquad\square$

Next, we provide an important lemma to ensure the concentration of the bonus term. The version for fixed $\phi$ is proved in Zanette et al. (2021a, Lemma 39). Here, we take a union bound over the whole feature $\phi \in \Phi$. Recall

$$
\rho_n(\cdot) = \frac{1}{n} \sum_{i=0}^{n-1} d_{P^\star}^{\pi_i}(\cdot).
$$

**Lemma 10** (Concentration of the bonus term). *Set $\lambda_n = \Theta(d \ln(n|\Phi|/\delta))$ for any $n$. Define*

$$
\Sigma_{\rho_n,\phi} = n\mathbb{E}_{s\sim\rho_n, a\sim U(\mathcal{A})}[\phi(s,a)\phi^\top(s,a)] + \lambda_n I, \quad \hat{\Sigma}_{n,\phi} = \sum_{i=0}^{n-1} \phi(s^{(i)}, a^{(i)})\phi^\top(s^{(i)}, a^{(i)}) + \lambda_n I.
$$

*With probability $1 - \delta$, we have*

$$
\forall n \in \mathbb{N}^+, \forall \phi \in \Phi, c_1\|\phi(s,a)\|_{\Sigma_{\rho_n \times U(\mathcal{A}),\phi}^{-1}} \leq \|\phi(s,a)\|_{\hat{\Sigma}_{n,\phi}^{-1}} \leq c_2\|\phi(s,a)\|_{\Sigma_{\rho_n \times U(\mathcal{A}),\phi}^{-1}}.
$$

For any $g \in \mathcal{S} \times \mathcal{A} \to \mathbb{R}$, The next lemma shows that $\mathbb{E}_{(s,a) \sim d^{\pi}_{\hat{P}_n}} \{g(s,a)\}$ can be upper-bounded using $\mathbb{E}_{(\tilde{s},\tilde{a}) \sim d^{\pi}_{\hat{P}_n}} \|\hat{\phi}_n(\tilde{s},\tilde{a})\|_{\Sigma^{-1}_{\rho_n,\hat{\phi}_n}}$ as long as we have the convergence guarantee for

$$\mathbb{E}_{s \sim \rho_n, a \sim U(\mathcal{A})} \left[g^2(s,a)\right] \text{ and } \mathbb{E}_{s \sim \rho'_n, a \sim U(\mathcal{A})} \left[g^2(s,a)\right].$$

**Lemma 11** (One-step back inequality for the learned model). *Take any $g \in \mathcal{S} \times \mathcal{A} \to \mathbb{R}$ such that $\|g\|_\infty \leq B$. We condition on the event where the MLE guarantee (17):*

$$\mathbb{E}_{s \sim \rho_n, a \sim U(\mathcal{A})}[f_n(s,a)] \lesssim \zeta_n,$$

*holds. Then, for any policy $\pi$, we have*

$$|\mathbb{E}_{(s,a) \sim d^{\pi}_{\hat{P}_n}} \{g(s,a)\}|$$

$$\leq \mathbb{E}_{(\tilde{s},\tilde{a}) \sim d^{\pi}_{\hat{P}_n}} \|\hat{\phi}_n(\tilde{s},\tilde{a})\|_{\Sigma^{-1}_{\rho_n \times U(\mathcal{A}), \hat{\phi}_n}} \sqrt{\left\{n|\mathcal{A}|\mathbb{E}_{s \sim \rho'_n, a \sim U(\mathcal{A})} \left[g^2(s,a)\right]\right\} + B^2 \lambda_n d + n B^2 \zeta_n}$$

$$+ \sqrt{(1-\gamma)|\mathcal{A}|\mathbb{E}_{s \sim \rho_n, a \sim U(\mathcal{A})} \left[g^2(s,a)\right]}.$$

Recall $\Sigma_{\rho_n \times U(\mathcal{A}), \hat{\phi}_n} = n\mathbb{E}_{s \sim \rho_n, a \sim U(\mathcal{A})}[\hat{\phi}_n(s,a)\hat{\phi}_n^\top(s,a)] + \lambda_n I$.

*Proof.* First, we have an equality:

$$\mathbb{E}_{(s,a) \sim d^{\pi}_{\hat{P}_n}} \{g(s,a)\} = \gamma \mathbb{E}_{(\tilde{s},\tilde{a}) \sim d^{\pi}_{\hat{P}_n}, s \sim \hat{P}_n(\tilde{s},\tilde{a}), a \sim \pi(s)} \{g(s,a)\} + (1-\gamma)\mathbb{E}_{s \sim d_0, a \sim \pi(s_0)} \{g(s,a)\}, \tag{8}$$

The second term in (8) is upper-bounded by

$$(1-\gamma)\sqrt{\max_{(s,a)} \frac{d_0(s)\pi(a \mid s)}{\rho_n(s)u(a)} \mathbb{E}_{s \sim \rho_n, a \sim U(\mathcal{A})} \left[g^2(s,a)\right]}$$

$$\leq (1-\gamma)\sqrt{\max_{(s,a)} \frac{d_0(s)\pi(a \mid s)}{(1-\gamma)d_0(s)u(a)} \mathbb{E}_{s \sim \rho_n, a \sim U(\mathcal{A})} \left[g^2(s,a)\right]} \leq \sqrt{(1-\gamma)|\mathcal{A}|\mathbb{E}_{s \sim \rho_n, a \sim U(\mathcal{A})} \left[g^2(s,a)\right]}.$$

Next we consider the first term in (8). By CS inequality, we have

$$\mathbb{E}_{(\tilde{s},\tilde{a}) \sim d^{\pi}_{\hat{P}_n}, s \sim \hat{P}_n(\tilde{s},\tilde{a}), a \sim \pi(s)} \{g(s,a)\} = \mathbb{E}_{(\tilde{s},\tilde{a}) \sim d^{\pi}_{\hat{P}_n}} \hat{\phi}_n(\tilde{s},\tilde{a})^\top \int \sum_a \hat{\mu}_n(s)\pi(a \mid s)g(s,a)d(s)$$

$$\leq \mathbb{E}_{(\tilde{s},\tilde{a}) \sim d^{\pi}_{\hat{P}_n}} \|\hat{\phi}_n(\tilde{s},\tilde{a})\|_{\Sigma^{-1}_{\rho_n \times U(\mathcal{A}), \hat{\phi}_n}} \left\| \int \sum_a \hat{\mu}_n(s)\pi(a \mid s)g(s,a)d(s) \right\|_{\Sigma_{\rho_n \times U(\mathcal{A}), \hat{\phi}_n}}.$$

Then,

$$\left\| \int \sum_a \hat{\mu}_n(s)\pi(a \mid s)g(s,a)d(s) \right\|^2_{\Sigma_{\rho_n \times U(\mathcal{A}), \hat{\phi}_n}}$$

$$\leq \left\{ \int \sum_a \hat{\mu}_n(s)\pi(a \mid s)g(s,a)d(s) \right\}^\top \left\{ n\mathbb{E}_{s \sim \rho_n, a \sim U(\mathcal{A})}[\hat{\phi}_n \hat{\phi}_n^\top] + \lambda_n I \right\} \left\{ \int \sum_a \hat{\mu}_n(s)\pi(a \mid s)g(s,a)d(s) \right\}$$

$$\leq n\mathbb{E}_{\tilde{s} \sim \rho_n, \tilde{a} \sim U(\mathcal{A})} \left\{ \left[ \int \sum_a \hat{\mu}_n(s)^\top \hat{\phi}_n(\tilde{s},\tilde{a})\pi(a \mid s)g(s,a)d(s) \right]^2 \right\} + B^2 \lambda_n d$$

(Use the assumption $\|\sum_a \pi(a \mid s)g(s,a)\|_\infty \leq B$ and $\int \|\hat{\mu}_n(s)h(s)d(s)\|_2 \leq \sqrt{d}$ for any $h : \mathcal{S} \to [0,1]$.)

$$= n\mathbb{E}_{\tilde{s} \sim \rho_n, \tilde{a} \sim U(\mathcal{A})} \left[ \left\{ \mathbb{E}_{s \sim \hat{P}_n(\tilde{s},\tilde{a}), a \sim \pi(s)} [g(s,a)] \right\}^2 \right] + B^2 \lambda_n d$$

$$\leq n\mathbb{E}_{s \sim \rho_n, a \sim U(\mathcal{A})} \left[ \left\{ \mathbb{E}_{s \sim P^\star(\tilde{s},\tilde{a}), a \sim \pi(s)} [g(s,a)] \right\}^2 \right] + B^2 \lambda_n d + n B^2 \zeta_n \qquad \text{(MLE guarantee)}$$

$$\leq n\mathbb{E}_{\tilde{s} \sim \rho_n, \tilde{a} \sim U(\mathcal{A}), s \sim P^\star(\tilde{s},\tilde{a}), a \sim \pi(s)} \left[g^2(s,a)\right] + B^2 \lambda_n d + B^2 n\zeta_n. \qquad \text{(Jensen)}$$

$$\leq n|\mathcal{A}| \left\{ \mathbb{E}_{\tilde{s} \sim \rho_n, \tilde{a} \sim U(\mathcal{A}), s \sim P^\star(\tilde{s},\tilde{a}), a \sim U(\mathcal{A})} \left[g^2(s,a)\right] \right\} + B^2 \lambda_n d + B^2 n\zeta_n$$

(Importance sampling)

$$\leq n|\mathcal{A}|\mathbb{E}_{s \sim \rho'_n, a \sim U(\mathcal{A})} \left[g^2(s,a)\right] + B^2 \lambda_n d + B^2 n\zeta_n. \qquad \text{(Definition of } \rho'_n)$$

Then, the final statement is immediately concluded. $\qquad\square$

Below, we show a similar lemma as Lemma 11. The difference is we aim for calculating $\mathbb{E}_{(s,a)\sim d_{P^\star}^\pi}\{g(s,a)\}$ instead of $\mathbb{E}_{(s,a)\sim d_{\hat P_n}^\pi}\{g(s,a)\}$. For any $g\in\mathcal{S}\times\mathcal{A}\to\mathbb{R}$, this lemma shows that $\mathbb{E}_{(s,a)\sim d_{P^\star}^\pi}\{g(s,a)\}$ can be upper-bounded using $\mathbb{E}_{(\tilde s,\tilde a)\sim d_{P^\star}^\pi}\|\phi^\star(\tilde s,\tilde a)\|_{\Sigma^{-1}_{\rho_n,\hat\phi^\star}}$ as long as we have the convergence guarantee for $\mathbb{E}_{s\sim\rho_n,a\sim U(\mathcal{A})}[g^2(s,a)]$. Note comparing to Lemma 11, this is not a probabilistic statement. Note that $\|\phi^\star(s,a)\|_{\Sigma^{-1}_{\rho_n,\phi^\star}}$ is the usual elliptical potential function under the fixed representation $\phi^\star$.

**Lemma 12** (One-step back inequality for the true model ). *Take any $g\in\mathcal{S}\times\mathcal{A}\to\mathbb{R}$ such that $\|g\|_\infty\le B$. Then,*

$$\mathbb{E}_{(s,a)\sim d_{P^\star}^\pi}\{g(s,a)\}\le\mathbb{E}_{(\tilde s,\tilde a)\sim d_{P^\star}^\pi}\|\phi^\star(\tilde s,\tilde a)\|_{\Sigma^{-1}_{\rho_n,\phi^\star}}\sqrt\gamma\sqrt{n|\mathcal{A}|\mathbb{E}_{s\sim\rho_n,a\sim U(\mathcal{A})}[g^2(s,a)]+\lambda_n dB^2}$$
$$+\sqrt{(1-\gamma)|\mathcal{A}|\mathbb{E}_{s\sim\rho_n,a\sim U(\mathcal{A})}[g^2(s,a)]}.$$

Recall $\Sigma_{\rho_n,\phi^\star}=n\mathbb{E}_{(s,a)\sim\rho_n}[\phi^\star(s,a)\phi^\star(s,a)^\top]+\lambda_n I$.

*Proof.* First, we have

$$\mathbb{E}_{(s,a)\sim d_{P^\star}^\pi}\{g(s,a)\}=\gamma\mathbb{E}_{(\tilde s,\tilde a)\sim d_{P^\star}^\pi,s\sim P^\star(\tilde s,\tilde a),a\sim\pi(s)}\{g(s,a)\}+(1-\gamma)\mathbb{E}_{s\sim d_0,a\sim\pi(s_0)}\{g(s,a)\}. \tag{9}$$

The second term in (9) is upper-bounded by

$$(1-\gamma)\sqrt{\max_{(s,a)}\frac{d_0(s)\pi(a\mid s)}{\rho_n(s)u(a)}\mathbb{E}_{s\sim\rho_n,a\sim U(\mathcal{A})}[g^2(s,a)]}\le\sqrt{|\mathcal{A}|\mathbb{E}_{s\sim\rho_n,a\sim U(\mathcal{A})}[g^2(s,a)]\,(1-\gamma)}.$$

By CS inequality, the first term in (9) is further bounded as follows:

$$\mathbb{E}_{(\tilde s,\tilde a)\sim d_{P^\star}^\pi,s\sim P^\star(\tilde s,\tilde a),a\sim\pi(s)}\{g(s,a)\}=\mathbb{E}_{(\tilde s,\tilde a)\sim d_{P^\star}^\pi}\phi^\star(\tilde s,\tilde a)^\top\int\sum_a\mu^\star(s)\pi(a\mid s)g(s,a)d(s)$$

$$\le\mathbb{E}_{(\tilde s,\tilde a)\sim d_{P^\star}^\pi}\|\phi^\star(\tilde s,\tilde a)\|_{\Sigma^{-1}_{\rho_n,\phi^\star}}\left\|\int\sum_a\mu^\star(s)\pi(a\mid s)g(s,a)d(s)\right\|_{\Sigma_{\rho_n,\phi^\star}}.$$

Here, we have

$$\|\int\sum_a\mu^\star(s)\pi(a\mid s)g(s,a)d(s)\|^2_{\Sigma_{\rho_n,\phi^\star}}$$

$$\le\left\{\int\sum_a\mu^\star(s)\pi(a\mid s)g(s,a)d(s)\right\}^\top\{n\mathbb{E}_{(s,a)\sim\rho_n}[\phi^\star(s,a)\{\phi^\star(s,a)\}^\top]+\lambda_n I\}\left\{\int\sum_a\mu^\star(s)\pi(a\mid s)g(s,a)d(s)\right\}$$

$$\le n\mathbb{E}_{(\tilde s,\tilde a)\sim\rho_n}\left\{\left[\int\sum_a\mu^\star(s)^\top\phi^\star(\tilde s,\tilde a)\pi(a\mid s)g(s,a)d(s)\right]^2\right\}+\lambda_n dB^2$$

$$\le n\left\{\mathbb{E}_{(\tilde s,\tilde a)\sim\rho_n,s\sim P^\star(\tilde s,\tilde a),a\sim\pi(s)}[g^2(s,a)]\right\}+\lambda_n dB^2. \tag{Jensen}$$

Therefore,

$$n\left\{\mathbb{E}_{(\tilde s,\tilde a)\sim\rho_n,s\sim P^\star(\tilde s,\tilde a),a\sim\pi(s)}[g^2(s,a)]\right\}+\lambda_n dB$$
$$\le n|\mathcal{A}|\left\{\mathbb{E}_{(\tilde s,\tilde a)\sim\rho_n,s\sim P^\star(\tilde s,\tilde a),a\sim U(\mathcal{A})}[g^2(s,a)]\right\}+\lambda_n dB^2 \qquad\text{(Importance sampling)}$$
$$\le n|\mathcal{A}|\left\{\frac{1}{\gamma}\mathbb{E}_{s\sim\rho_n,a\sim U(\mathcal{A})}[g^2(s,a)]\right\}+\lambda_n dB^2.$$

In the last line, we use the following inequality:

$$\mathbb{E}_{s\sim\rho_n,a\sim U(\mathcal{A})}\left[g^2(s,a)\right]$$
$$= \gamma\mathbb{E}_{(\tilde{s},\tilde{a})\sim\rho_n,s\sim P^\star(\tilde{s},\tilde{a}),a\sim U(\mathcal{A})}\left[g^2(s,a)\right] + (1-\gamma)\mathbb{E}_{s_0\sim d_0,a\sim U(\mathcal{A})}\left[g^2(s,a)\right]$$
$$\geq \gamma\mathbb{E}_{(\tilde{s},\tilde{a})\sim\rho_n,s\sim P^\star(\tilde{s},\tilde{a}),a\sim U(\mathcal{A})}\left[g^2(s,a)\right].$$

Finally, we have

$$\mathbb{E}_{(s,a)\sim d_{P^\star}^\pi}\left\{g(s,a)\right\} \leq \mathbb{E}_{(\tilde{s},\tilde{a})\sim d_{P^\star}^\pi}\|\phi^\star(\tilde{s},\tilde{a})\|_{\Sigma_{\rho_n,\phi^\star}^{-1}}\sqrt{\gamma}\sqrt{\left\{n|\mathcal{A}|\mathbb{E}_{s\sim\rho_n,a\sim U(\mathcal{A})}\left[g^2(s,a)\right]\right\} + \lambda_n dB^2}$$
$$+ \sqrt{|\mathcal{A}|\mathbb{E}_{s\sim\rho_n,a\sim U(\mathcal{A})}\left[g^2(s,a)\right](1-\gamma)}.$$

This concludes the proof. $\qquad\square$

## C  PROOF OF THE THEORETICAL PROPERTY OF REP-LCB

This section provides the detailed proofs for our results in the offline setting.

Below we first prove that $V_{\hat{P},r-\hat{b}}^\pi$ is an almost pessimistic estimator of $V_{P^\star,r}^\pi$.

**Lemma 13** (Almost Pessimism at the Initial Distribution). *Let $\omega = \max_{a,s} 1/\pi_b(a\mid s)$. Set*

$$\alpha = c_1\sqrt{(\omega+d^2)\gamma\ln(|\mathcal{M}|/\delta)}, \quad \lambda = O(d\ln(|\mathcal{M}|/\delta)), \zeta = O\left(\frac{\ln(|\mathcal{M}|/\delta)}{n}\right).$$

*With probability $1-\delta$, for any policy $\pi$, we have*

$$V_{\hat{P},r-\hat{b}}^\pi - V_{P^\star,r}^\pi \leq \sqrt{\frac{\omega(1-\gamma)^{-1}\ln(|\mathcal{M}|/\delta)}{n}}.$$

*Proof.* We define

$$\Sigma_{\rho,\phi} = n\mathbb{E}_{(s,a)\sim\rho}[\phi\phi^\top] + \lambda I, \quad \hat{\Sigma}_\phi = n\mathbb{E}_{\mathcal{D}}[\phi\phi^\top] + \lambda I.$$

where $\lambda = O(d\ln(|\mathcal{M}|/\delta))$. In this proof, letting $f(s,a) = \|\hat{P}(\cdot\mid s,a) - P^\star(\cdot\mid s,a)\|_1$, we condition on the events:

$$\mathbb{E}_{(s,a)\sim\rho}[f^2(s,a)] \leq \zeta, \quad \forall\phi\in\Phi: \|\phi(s,a)\|_{\hat{\Sigma}_\phi^{-1}} = \Theta(\|\phi(s,a)\|_{\Sigma_{\rho,\phi}^{-1}}). \tag{10}$$

where $\zeta = O(\ln(|\mathcal{M}|/\delta)/n)$. From the offline version of Lemma 17 and Lemma 10 [6], this event happens with probability $1-\delta$.

Then, from simulation lemma (Lemma 20),

$$(1-\gamma)(V_{\hat{P},r-\hat{b}}^\pi - V_{P^\star,r}^\pi)$$
$$= \mathbb{E}_{(s,a)\sim d_{\hat{P}}^\pi}\left[-\hat{b}(s,a) + \gamma\mathbb{E}_{s'\sim\hat{P}(s,a)}\left[V_{P^\star,r}^\pi(s')\right] - \gamma\mathbb{E}_{s'\sim P^\star(s,a)}\left[V_{P^\star,r}^\pi(s')\right]\right]$$
$$\lesssim \mathbb{E}_{(s,a)\sim d_{\hat{P}}^\pi}\left[-\min\left(\alpha\|\hat{\phi}(s,a)\|_{\Sigma_{\rho,\hat{\phi}}^{-1}},2\right) + \gamma\mathbb{E}_{s'\sim\hat{P}(s,a)}\left[V_{P^\star,r}^\pi(s')\right] - \gamma\mathbb{E}_{s'\sim P^\star(s,a)}\left[V_{P^\star,r}^\pi(s')\right]\right].$$
(From (10))

Here, we have

$$\left|\mathbb{E}_{(s,a)\sim d_{\hat{P}}^\pi}\left\{\mathbb{E}_{s'\sim\hat{P}(s,a)}\left[V_{P^\star,r}^\pi(s')\right] - \mathbb{E}_{s'\sim P^\star(s,a)}\left[V_{P^\star,r}^\pi(s')\right]\right\}\right| \leq \mathbb{E}_{(s,a)\sim d_{\hat{P}}^\pi}\left\{f(s,a)\right\},$$

noting $\|V_{P^\star,r}^\pi\|_\infty \leq 1$. This is further bounded by Lemma 15:

$$\mathbb{E}_{(s,a)\sim d_{\hat{P}}^\pi}\left\{f(s,a)\right\} \lesssim \sqrt{\alpha'}\mathbb{E}_{(\tilde{s},\tilde{a})\sim d_{\hat{P}}^\pi}\|\hat{\phi}(\tilde{s},\tilde{a})\|_{\Sigma_{\rho,\hat{\phi}}^{-1}} + \sqrt{\omega\zeta(1-\gamma)}. \tag{11}$$

---

[6]We can remove $\ln n$ since $n$ is fixed in the offline setting.

where

$$\alpha' = n\gamma\omega\zeta + \gamma^2\lambda d + \gamma^2 n\zeta \lesssim \left(\omega + d^2\right)\gamma\ln(|\mathcal{M}|/\delta).$$

Here, we use $f(s,a) \leq 2$ in Lemma 15 and $\mathbb{E}_{(s,a)\sim\rho}[f^2(s,a)] \leq \zeta$.

Thus,

$$
\begin{aligned}
&\left|\mathbb{E}_{(s,a)\sim d_{\hat{P}}^{\pi}}\left\{\mathbb{E}_{s'\sim\hat{P}(s,a)}\left[V_{P^{\star},r}^{\pi}(s')\right] - \mathbb{E}_{s'\sim P^{\star}(s,a)}\left[V_{P^{\star},r}^{\pi}(s')\right]\right\}\right| \leq \mathbb{E}_{(s,a)\sim d_{\hat{P}}^{\pi}}\left\{f(s,a)\right\}\\
&\leq \sqrt{\alpha'}\mathbb{E}_{(\tilde{s},\tilde{a})\sim d_{\hat{P}}^{\pi}}\|\hat{\phi}(\tilde{s},\tilde{a})\|_{\Sigma_{\rho,\hat{\phi}}^{-1}} + \sqrt{\omega\zeta(1-\gamma)}\\
&= \alpha\mathbb{E}_{(\tilde{s},\tilde{a})\sim d_{\hat{P}}^{\pi}}\|\hat{\phi}(\tilde{s},\tilde{a})\|_{\Sigma_{\rho,\hat{\phi}}^{-1}} + \sqrt{\omega\zeta(1-\gamma)}, \quad \alpha = \sqrt{\alpha'}.
\end{aligned}
$$

Going back to the simulation lemma 20, we have

$$
\begin{aligned}
&(1-\gamma)(V_{\hat{P},r-\hat{b}}^{\pi} - V_{P^{\star},r}^{\pi})\\
&\lesssim \mathbb{E}_{(s,a)\sim d_{\hat{P}}^{\pi}}\left[-\min\left(\alpha\|\hat{\phi}(s,a)\|_{\Sigma_{\rho,\hat{\phi}}^{-1}}, 2\right) + \mathbb{E}_{s'\sim\hat{P}(s,a)}\left[V_{P^{\star},r}^{\pi}(s')\right] - \mathbb{E}_{s'\sim P^{\star}(s,a)}\left[V_{P^{\star},r}^{\pi}(s')\right]\right]\\
&\leq \mathbb{E}_{(s,a)\sim d_{\hat{P}}^{\pi}}\left[-\min\left(\alpha\|\hat{\phi}(s,a)\|_{\Sigma_{\rho,\hat{\phi}}^{-1}}, 2\right) + \min\left(\alpha\|\hat{\phi}(s,a)\|_{\Sigma_{\rho,\hat{\phi}}^{-1}} + \sqrt{\omega\zeta(1-\gamma)}, 2\right)\right]\\
&\leq \sqrt{\omega\zeta(1-\gamma)}.
\end{aligned}
$$

This concludes the proof. $\qquad\square$

With the above lemma, now we can proceed to prove the main theorem.

**Theorem 14** (PAC guarantee of REP-LCB). *Set the parameters as in Lemma 13. With probability $1-\delta$, for any comparator policy $\pi$ including history-dependent non-Markovian policies, we have*

$$V_{P^{\star},r}^{\pi} - V_{P^{\star},r}^{\hat{\pi}} \lesssim \frac{\omega d^2}{(1-\gamma)^2}\sqrt{\frac{C_{\pi}^{\star}\ln(|\mathcal{M}|/\delta)}{n}},$$

*where $C_{\pi}^{\star}$ is the relative condition number under $\phi^{\star}$:*

$$C_{\pi}^{\star} := \sup_{x\in\mathbb{R}}\frac{x^{\top}\mathbb{E}_{(s,a)\sim d_{P^{\star}}^{\pi}}[\phi^{\star}(s,a)\{\phi^{\star}(s,a)\}^{\top}]x}{x^{\top}\mathbb{E}_{(s,a)\sim\rho}[\phi^{\star}(s,a)\{\phi^{\star}(s,a)\}^{\top}]x}.$$

*Proof.* In this proof, letting $f(s,a) = \|\hat{P}(\cdot\mid s,a) - P^{\star}(\cdot\mid s,a)\|_1$ we condition on the events:

$$\mathbb{E}_{(s,a)\sim\rho}[f^2(s,a)] \leq \zeta, \quad \forall\phi\in\Phi : \|\phi(s,a)\|_{\hat{\Sigma}_{\phi}^{-1}} = \Theta(\|\phi(s,a)\|_{\Sigma_{\rho,\phi}^{-1}}). \tag{12}$$

From Lemma 10 and Lemma 17, this event happens with probability $1-\delta$.

For any policy $\pi$, we have

$$
\begin{aligned}
&V_{P^{\star},r}^{\pi} - V_{P^{\star},r}^{\hat{\pi}}\\
&\leq V_{P^{\star},r}^{\pi} - V_{\hat{P},r-\hat{b}}^{\hat{\pi}} + \sqrt{\omega\zeta(1-\gamma)^{-1}} && \text{(Lemma 13)}\\
&\leq V_{P^{\star},r}^{\pi} - V_{\hat{P},r-\hat{b}}^{\pi} + \sqrt{\omega\zeta(1-\gamma)^{-1}}\\
&\lesssim (1-\gamma)^{-1}\underbrace{\mathbb{E}_{(s,a)\sim d_{P^{\star}}^{\pi}}[\hat{b}(s,a)]}_{(a)} + \left(\frac{1}{1-\gamma}\right)^2\underbrace{\mathbb{E}_{(s,a)\sim d_{P^{\star}}^{\pi}}[f(s,a)]}_{(b)} + \sqrt{\omega\zeta(1-\gamma)^{-1}}.
\end{aligned}
$$

Recall $f(s,a) = \|\hat{P}(\cdot\mid s,a) - P^{\star}(\cdot\mid s,a)\|_1$.

From the second line to the third line, note though $\hat{\pi}$ is the argmax over Markovian polices, $\hat{\pi}$ is also the argmax over all history-dependent polices. In the last line, we use a simulation lemma 20, which is tailored to a time-inhomogeneous policy. We here use $\|V_{\hat{P},r-\hat{b}}^{\pi}\|_{\infty} \leq 2/((1-\gamma))$. noting $\|\hat{b}\|_{\infty} = O(1)$.

We further calculate the first term (a). Considering 16 and noting $\|\hat{b}\|_\infty \le 2$, we have

$$\mathbb{E}_{(s,a)\sim d^\pi_{P^\star}}[\hat{b}(s,a)] \lesssim \mathbb{E}_{(\tilde{s},\tilde{a})\sim d^\pi_{P^\star}}\|\phi^\star(\tilde{s},\tilde{a})\|_{\Sigma^{-1}_{\rho,\phi^\star}}\sqrt{n\omega\left\{\gamma\mathbb{E}_{(s,a)\sim\rho}\left[\hat{b}^2(s,a)\right]\right\}+\gamma\lambda d}$$
$$+\sqrt{\omega(1-\gamma)}\{\mathbb{E}_\rho[\hat{b}^2(s,a)]\}^{1/2}.$$

From (12), we have

$$n\mathbb{E}_{(s,a)\sim\rho}\left[\hat{b}^2(s,a)\right] \le n\mathbb{E}_{(s,a)\sim\rho}\left[\min\left(\alpha^2\|\hat{\phi}(s,a)\|^2_{\Sigma^{-1}_{\rho,\hat\phi}},4\right)\right] \le n\mathbb{E}_{(s,a)\sim\rho}\left[\alpha^2\|\hat{\phi}(s,a)\|^2_{\Sigma^{-1}_{\rho,\hat\phi}}\right] \tag{13}$$

$$\le \mathrm{Tr}[n\mathbb{E}_{(s,a)\sim\rho}[\hat\phi\hat\phi^\top]\{n\mathbb{E}_{(s,a)\sim\rho}[\hat\phi\hat\phi^\top]+\lambda I\}^{-1} \tag{14}$$
$$\le \mathrm{Tr}[n(\mathbb{E}_{(s,a)\sim\rho}[\hat\phi\hat\phi^\top]+\lambda I)\{n\mathbb{E}_{(s,a)\sim\rho}[\hat\phi\hat\phi^\top]+\lambda I\}^{-1}] \le d. \tag{15}$$

Thus,

$$\mathbb{E}_{(s,a)\sim d^\pi_{P^\star}}[\hat{b}(s,a)] \le \mathbb{E}_{(\tilde{s},\tilde{a})\sim d^\pi_{P^\star}}\|\phi^\star(\tilde{s},\tilde{a})\|_{\Sigma^{-1}_{\rho,\phi^\star}}\sqrt{\omega d\alpha^2\gamma+\gamma\lambda d}+\sqrt{\frac{\omega d\alpha^2(1-\gamma)}{n}}.$$

Second, we further calculate the second term (b). Considering the offline version of Lemma 12 and noting $f^2(s,a)$ is upper-bounded by $4$,

$$\mathbb{E}_{(s,a)\sim d^\pi_{P^\star}}[f(s,a)]$$

$$=\mathbb{E}_{(\tilde{s},\tilde{a})\sim d^\pi_{P^\star}}\|\phi^\star(\tilde{s},\tilde{a})\|_{\Sigma^{-1}_{\rho,\phi^\star}}\sqrt{n\omega\left\{\gamma\mathbb{E}_{(s,a)\sim\rho}[f^2(s,a)]\right\}+4\gamma\lambda d}+\sqrt{\omega\mathbb{E}_{(s,a)\sim\rho}[f^2(s,a)](1-\gamma)}$$

$$\lesssim \mathbb{E}_{(\tilde{s},\tilde{a})\sim d^\pi_{P^\star}}\|\phi^\star(\tilde{s},\tilde{a})\|_{\Sigma^{-1}_{\rho,\phi^\star}}\sqrt{\omega\{n\gamma\zeta\}+\gamma\lambda d}+\sqrt{\omega\zeta(1-\gamma)}$$

$$\lesssim \mathbb{E}_{(\tilde{s},\tilde{a})\sim d^\pi_{P^\star}}\|\phi^\star(\tilde{s},\tilde{a})\|_{\Sigma^{-1}_{\rho,\phi^\star}}\alpha+\sqrt{\omega\zeta(1-\gamma)}.$$

In the final line, recall $\sqrt{\omega\{n\gamma\zeta\}+\gamma\lambda d+\gamma n\zeta} \le \alpha$.

Finally, by combining the calculation of the first term (a) and the second term (b), we have

$$V^\pi_{P^\star,r} - V^{\hat\pi}_{P^\star,r} \lesssim \frac{1}{(1-\gamma)}\mathbb{E}_{(\tilde{s},\tilde{a})\sim d^\pi_{P^\star}}\|\phi^\star(\tilde{s},\tilde{a})\|_{\Sigma^{-1}_{\rho,\phi^\star}}\sqrt{d\alpha^2\omega\gamma+\gamma\lambda d}+\sqrt{\frac{\omega\alpha^2 d(1-\gamma)^{-1}}{n}}$$

$$+\frac{\alpha}{(1-\gamma)^2}\mathbb{E}_{(\tilde{s},\tilde{a})\sim d^\pi_{P^\star}}\|\phi^\star(\tilde{s},\tilde{a})\|_{\Sigma^{-1}_{\rho,\phi^\star}}+\sqrt{\frac{\omega\zeta}{(1-\gamma)^3}}$$

Now, we use the fact $\mathbb{E}_{(\tilde{s},\tilde{a})\sim d^\pi_{P^\star}}\|\phi^\star(\tilde{s},\tilde{a})\|_{\Sigma^{-1}_{\rho,\phi^\star}}$ is upper-bounded as

$$\mathbb{E}_{(\tilde{s},\tilde{a})\sim d^\pi_{P^\star}}\|\phi^\star(\tilde{s},\tilde{a})\|_{\Sigma^{-1}_{\rho,\phi^\star}} \le \sqrt{\mathbb{E}_{(\tilde{s},\tilde{a})\sim d^\pi_{P^\star}}\|\phi^\star(\tilde{s},\tilde{a})\|^2_{\Sigma^{-1}_{\rho,\phi^\star}}} \le \sqrt{C^\star\mathbb{E}_{(\tilde{s},\tilde{a})\sim\rho}\|\phi^\star(\tilde{s},\tilde{a})\|^2_{\Sigma^{-1}_{\rho,\phi^\star}}}$$

$$\text{(Refer to Lemma 21)}$$

$$\le \sqrt{C^\star d/n}. \qquad\qquad\qquad\qquad\text{(From (13))}$$

Finally, we have

$$V^\pi_{P^\star,r} - V^{\hat\pi}_{P^\star,r}$$

$$\lesssim (1-\gamma)^{-1}\left\{\sqrt{\frac{C^\star d}{n}}\sqrt{d\alpha^2\omega\gamma+\gamma\lambda d}+\frac{\alpha}{(1-\gamma)}\sqrt{\frac{C^\star d}{n}}+\sqrt{\frac{\omega\alpha^2 d(1-\gamma)}{n}}+\sqrt{\frac{\omega\zeta}{(1-\gamma)}}\right\}$$

$$\lesssim (1-\gamma)^{-1}\left\{\sqrt{\frac{C^\star d}{n}}\sqrt{d\alpha^2\omega\gamma}+\frac{\alpha}{(1-\gamma)}\sqrt{\frac{C^\star d}{n}}\right\} \qquad\text{(Take out two dominating terms)}$$

$$\lesssim \frac{\omega d^2}{(1-\gamma)^2}\sqrt{\frac{C^\star\ln(|\mathcal{M}|/\delta)}{n}}.$$

$\square$

The lemma below is a key technical lemma for our proof. It shows that one can relate the expected value of any function $f(s,a)$ with respect to $d_{\hat{P}}^\pi$ (i.e., inside the learned model $\hat{P}$) to the potential function with respect to $d_{\hat{P}}^\pi$, i.e., $\mathbb{E}_{(\tilde{s},\tilde{a})\sim d_{\hat{P}}^\pi}\|\hat{\phi}(\tilde{s},\tilde{a})\|_{\Sigma_{\rho,\hat{\phi}}^{-1}}$. Pairing $\hat{\phi}$ and $\hat{P}$ is important since $\hat{P}$ is the low-rank transition model defined using $\hat{\phi}$. As we have seen in the above analysis, when using the lemma below, we instantiate $f(s,a) := \|\hat{P}(\cdot|s,a) - P^\star(\cdot|s,a)\|_1$.

**Lemma 15** (One-step back inequality for the learned model in offline setting). *Take any $f \subset \mathcal{S} \times \mathcal{A} \to \mathbb{R}$ s.t. $\|f\|_\infty \le B$. We condition on the event where the MLE guarantee holds:*

$$\mathbb{E}_{(s,a)\sim\rho}\|\hat{P}(\cdot \mid s,a) - P^\star(\cdot \mid s,a)\|_1^2 \lesssim \zeta.$$

*Then, letting $\omega = \max_{s,a}(1/\pi_b(a \mid s))$, for any policy $\pi$, we have*

$$|\mathbb{E}_{(s,a)\sim d_{\hat{P}}^\pi}\{f(s,a)\}| \le \mathbb{E}_{(\tilde{s},\tilde{a})\sim d_{\hat{P}}^\pi}\|\hat{\phi}(\tilde{s},\tilde{a})\|_{\Sigma_{\rho,\hat{\phi}}^{-1}}\sqrt{\{n\omega\mathbb{E}_{(s,a)\sim\rho}[f^2(s,a)]\} + \gamma^2\lambda dB^2 + n\gamma^2\zeta B^2}$$
$$+ \sqrt{\omega\mathbb{E}_{(s,a)\sim\rho}[f^2(s,a)](1-\gamma)}.$$

*Proof.* First, we have an equality:

$$\mathbb{E}_{(s,a)\sim d_{\hat{P}}^\pi}\{f(s,a)\} = \gamma\mathbb{E}_{(\tilde{s},\tilde{a})\sim d_{\hat{P}}^\pi, s\sim\hat{P}(\tilde{s},\tilde{a}), a\sim\pi(s)}\{f(s,a)\} + (1-\gamma)\mathbb{E}_{s\sim d_0, a\sim\pi(s_0)}\{f(s,a)\}.$$
$$(16)$$

The second term in (16) is upper-bounded by

$$\mathbb{E}_{s\sim d_0, a\sim\pi(s_0)}\{f(s,a)\} \le \mathbb{E}_{s\sim d_0, a\sim\pi(s_0)}\{f^2(s,a)\}\}^{1/2} = \sqrt{\omega\mathbb{E}_{(s,a)\sim\rho}[f^2(s,a)]/(1-\gamma)}.$$

Next we consider the first term in (16). By CS inequality, we have

$$\left|\mathbb{E}_{(\tilde{s},\tilde{a})\sim d_{\hat{P}}^\pi, s\sim\hat{P}(\tilde{s},\tilde{a}), a\sim\pi(s)}\{f(s,a)\}\right| = \left|\mathbb{E}_{(\tilde{s},\tilde{a})\sim d_{\hat{P}}^\pi}\hat{\phi}(\tilde{s},\tilde{a})^\top\int\sum_a\hat{\mu}(s)\pi(a \mid s)f(s,a)d(s)\right|$$

$$\le \mathbb{E}_{(\tilde{s},\tilde{a})\sim d_{\hat{P}}^\pi}\|\hat{\phi}(\tilde{s},\tilde{a})\|_{\Sigma_{\rho,\hat{\phi}}^{-1}}\|\int\sum_a\hat{\mu}(s)\pi(a \mid s)f(s,a)d(s)\|_{\Sigma_{\rho,\hat{\phi}}}.$$

Then,

$$\|\int\hat{\mu}(s)\pi(a \mid s)f(s,a)d(s,a)\|_{\Sigma_{\rho,\hat{\phi}}}^2$$

$$\le \left\{\int\sum_a\hat{\mu}(s)\pi(a \mid s)f(s,a)d(s)\right\}^\top\left\{n\mathbb{E}_{(s,a)\sim\rho}[\hat{\phi}\hat{\phi}^\top] + \lambda I\right\}\left\{\int\sum_a\hat{\mu}(s)\pi(a \mid s)f(s,a)d(s)\right\}$$

$$\le n\mathbb{E}_{(\tilde{s},\tilde{a})\sim\rho}\left\{\left[\int\sum_a\hat{\mu}(s)^\top\hat{\phi}(\tilde{s},\tilde{a})\pi(a \mid s)f(s,a)d(s)\right]^2\right\} + B^2\lambda d$$

(Use the assumption $\|\sum_a f(s,a)\|_\infty \le B$ and $\|\int\hat{\mu}(s)h(s)\mathrm{d}(s)\|_2 \le \sqrt{d}$ for $h:\mathcal{S}\to[0,1]$.)

$$= n\mathbb{E}_{(\tilde{s},\tilde{a})\sim\rho}\{\mathbb{E}_{s\sim\hat{P}(\tilde{s},\tilde{a}), a\sim\pi(s)}[f(s,a)]^2\} + B^2\lambda d$$

$$= n\mathbb{E}_{(\tilde{s},\tilde{a})\sim\rho}\{\mathbb{E}_{s\sim P^\star(\tilde{s},\tilde{a}), a\sim\pi(s)}[f(s,a)]^2\} + B^2\lambda d + nB^2\zeta$$

(MLE guarantee and $\|\mathbb{E}_{a\sim\pi(\cdot)}[f^2(\cdot,a)]\|_\infty \le B^2$.)

$$\le n\{\mathbb{E}_{(\tilde{s},\tilde{a})\sim\rho, s\sim P^\star(\tilde{s},\tilde{a}), a\sim\pi(s)}[f^2(s,a)]\} + B^2\lambda d + nB^2\zeta.$$
$\qquad$ (Jensen)

Finally, the first term in (16) is upper-bounded by

$$n\{\mathbb{E}_{(\tilde{s},\tilde{a})\sim\rho, s\sim P^\star(\tilde{s},\tilde{a}), a\sim\pi(s)}[f^2(s,a)]\} + \lambda dB^2 + nB^2\zeta$$

$$\le n\omega\{\mathbb{E}_{(\tilde{s},\tilde{a})\sim\rho, s\sim P^\star(\tilde{s},\tilde{a}), a\sim\pi_b(s)}[f^2(s,a)]\} + \lambda dB^2 + nB^2\zeta \qquad \text{(Importance sampling)}$$

$$\le n\omega\left\{\frac{1}{\gamma}\mathbb{E}_{(s,a)\sim\rho}[f^2(s,a)]\right\} + \lambda dB^2 + nB^2\zeta. \qquad \text{(Definition of } \rho\text{)}$$

In the last line, we use the following equality:

$$\mathbb{E}_{(s,a)\sim\rho}\left[f^2(s,a)\right] = \gamma\mathbb{E}_{(\tilde{s},\tilde{a})\sim\rho,s\sim P^\star(\tilde{s},\tilde{a}),a\sim\pi_b(s)}\left[f^2(s,a)\right] + (1-\gamma)\mathbb{E}_{s\sim d_0,a\sim\pi_b}\left[f^2(s,a)\right].$$

Based on the above discussion, the final statement is immediately concluded.

$\square$

We can prove the similar inequality for the true model. The proof is omitted since it is quite similar to the one of Lemma 15.

**Lemma 16** (One-step back inequality for the true model in offline setting). *Take any $f \subset \mathcal{S} \times \mathcal{A} \to \mathbb{R}$ s.t. $\|f\|_\infty \leq B$. Then, letting $\omega = \max_{s,a}(1/\pi_b(a \mid s))$, for any policy $\pi$, we have*

$$\left|\mathbb{E}_{(s,a)\sim d^\pi_{P^\star}}\left\{f(s,a)\right\}\right| \leq \mathbb{E}_{(\tilde{s},\tilde{a})\sim d^\pi_{P^\star}}\|\phi^\star(\tilde{s},\tilde{a})\|_{\Sigma^{-1}_{\rho,\phi^\star}}\sqrt{\left\{n\omega\mathbb{E}_{(s,a)\sim\rho}\left[f^2(s,a)\right]\right\} + \gamma^2\lambda d B^2}$$
$$+ \sqrt{\omega\mathbb{E}_{(s,a)\sim\rho}\left[f^2(s,a)\right](1-\gamma)}.$$

## D AUXILIARY LEMMAS

First, we present the MLE guarantee. Regarding the proof, refer to Agarwal et al. (2020b, Theorem 21). Note $\hat{P}_n$ and $\bar{\pi}_n$ are the quantities appearing in the proposed online algorithm. We can also immediately obtain the statement to the offline case.

**Lemma 17** (MLE guarantee). *For a fixed episode $n$, with probability $1 - \delta$,*

$$\mathbb{E}_{s\sim\{0.5\rho_n+0.5\rho'_n\},a\sim U(\mathcal{A})}[\|\hat{P}_n(\cdot \mid s,a) - P^\star(\cdot \mid s,a)\|_1^2] \lesssim \zeta, \quad \zeta := \frac{\ln(|\mathcal{M}|/\delta)}{n}.$$

*As a straightforward corollary, with probability $1 - \delta$,*

$$\forall n \in \mathbb{N}^+, \mathbb{E}_{s\sim\{0.5\rho_n+0.5\rho'_n\},a\sim U(\mathcal{A})}[\|\hat{P}_n(\cdot \mid s,a) - P^\star(\cdot \mid s,a)\|_1^2] \lesssim 0.5\zeta_n, \quad \zeta_n := \frac{\ln(|\mathcal{M}|n/\delta)}{n}. \tag{17}$$

The following is a standard inequality to prove regret bounds for linear models. Refer to Agarwal et al. (2020a, Lemma G.2.)

**Lemma 18.** *Consider the following process. For $n = 1, \cdots, N$, $M_n = M_{n-1}+G_n$ with $M_0 = \lambda_0 I$ and $G_n$ being a positive semidefinite matrix with eigenvalues upper-bounded by 1. We have that:*

$$2\ln\det(M_N) - 2\ln\det(\lambda_0 I) \geq \sum_{n=1}^N \text{Tr}(G_n M_{n-1}^{-1}).$$

**Lemma 19** (Potential function lemma). *Suppose $\text{Tr}(G_n) \leq B^2$.*

$$2\ln\det(M_N) - 2\ln\det(\lambda_0 I) \leq d\ln\left(1 + \frac{NB^2}{d\lambda_0}\right).$$

*Proof.* Let $\sigma_1, \cdots, \sigma_d$ be the set of singular values of $M_N$ recalling $M_N$ is a positive semidefinite matrix. Then, by the AM-GM inequality,

$$\ln\det(M_N)/\det(\lambda_0 I) = \ln\prod_{i=1}^d(\sigma_i/\lambda_0) \leq \ln d\left(\frac{1}{d}\sum_{i=1}^d(\sigma_i/\lambda_0))\right)$$

Since we have $\sum_i \sigma_i = \text{Tr}(M_N) \leq d\lambda_0 + NB^2$, the statement is concluded. $\square$

**Lemma 20** (Simulation lemma). *Given two MDPs $(P', r+b)$ and $(P, r)$, for any policy $\pi$, we have:*

$$V^\pi_{P',r+b} - V^\pi_{P,r} = \frac{1}{1-\gamma}\mathbb{E}_{(s,a)\sim d^\pi_{P'}}[b(s,a) + \gamma\mathbb{E}_{P'(s'|s,a)}[Q^\pi_{P,r}(s',\pi)] - \gamma\mathbb{E}_{P(s'|s,a)}[Q^\pi_{P,r}(s',\pi)]]$$

*and*

$$V^\pi_{P',r+b} - V^\pi_{P,r} = \frac{1}{1-\gamma}\mathbb{E}_{(s,a)\sim d^\pi_P}[b(s,a) + \gamma\mathbb{E}_{P'(s'|s,a)}[Q^\pi_{P,r+b}(s',\pi)] - \gamma\mathbb{E}_{P(s'|s,a)}[Q^\pi_{P',r+b}(s',\pi)]].$$

*Proof.* We use

$$V_P^\pi - f(s_0, \pi) = \frac{1}{1-\gamma}\mathbb{E}_{d_P^\pi}[r(s,a) + \gamma\mathbb{E}_{P(s'|s,a)}[f(s',\pi)] - f(s,a)]]$$

Then,

$$V_{P',r+b}^\pi - V_{P,r}^\pi = \frac{1}{1-\gamma}\mathbb{E}_{(s,a)\sim d_{P'}^\pi}[r(s,a) + b(s,a) + \gamma\mathbb{E}_{P'(s'|s,a)}[Q_{P,r}^\pi(s',\pi)] - Q_{P,r}^\pi(s,a)]]$$

$$= \frac{1}{1-\gamma}\mathbb{E}_{(s,a)\sim d_{P'}^\pi}[b(s,a) + \gamma\mathbb{E}_{P'(s'|s,a)}[Q_{P,r}^\pi(s',\pi)] - \gamma\mathbb{E}_{P(s'|s,a)}[Q_{P,r}^\pi(s',\pi)]].$$

Similarly,

$$V_{P,r}^\pi - V_{P',r+b}^\pi = \frac{1}{1-\gamma}\mathbb{E}_{(s,a)\sim d_P^\pi}[r(s,a) + \gamma\mathbb{E}_{P(s'|s,a)}[Q_{P',r+b}^\pi(s',\pi)] - Q_{P',r+b}^\pi(s,a)]]$$

$$= \frac{1}{1-\gamma}\mathbb{E}_{(s,a)\sim d_P^\pi}[-b(s,a) + \gamma\mathbb{E}_{P(s'|s,a)}[Q_{P',r+b}^\pi(s',\pi)] - \gamma\mathbb{E}_{P'(s'|s,a)}[Q_{P,r}^\pi(s',\pi)]].$$

$\square$

The following lemma is used to deal with the distribution shift in the offline setting. For the proof, refer to Chang et al. (2021).

**Lemma 21** (Distribution shift lemma). *Consider any policy $\pi$ and state-action distribution $\rho$, and any representation $\phi^\star$, we have:*

$$\mathbb{E}_{(s,a)\sim d_{P^\star}^\pi}[\phi^\star(s,a)\{\phi^\star(s,a)\}^\top] \leq C^\star\mathbb{E}_\rho[\phi^\star(s,a)\{\phi^\star(s,a)\}^\top], \quad C^\star := \sup_{x\in\mathbb{R}}\frac{x^\top\mathbb{E}_{(s,a)\sim d_{P^\star}^\pi}[\phi^\star\{\phi^\star\}^\top]x}{x^\top\mathbb{E}_{(s,a)\sim\rho}[\phi^\star\{\phi^\star\}^\top]x}.$$

This is some auxiliary lemma to convert the finite sample error bound into the sample complexity.

**Lemma 22** (Conversion of finite sample error bounds into sample complexities). *By taking*

$$N = 1/\epsilon'^2 \times \ln^2(1 + 1/\epsilon'^2), \epsilon' = \frac{\epsilon}{a_1 \ln^{1/2}(e + a_2)\ln^{1/2}(e + a_3)}.$$

*It satisfies*

$$a_1\sqrt{1/N}\ln^{1/2}(1 + a_2N)\ln^{1/2}(1 + a_3N) < c\epsilon.$$

*where $c$ is a constant independent of $a_1, a_2, a_3$.*

*Proof.* We first have

$$a_1\sqrt{1/N}\ln^{1/2}(1 + a_2N)\ln^{1/2}(1 + a_3N) \leq a_1\max(\ln^{1/2}(1 + a_2)\ln^{1/2}(1 + a_3), 1)\sqrt{1/N}\ln(1 + N).$$

Here, we use

$$\ln^{1/2}(1 + a_2N) \leq \{\ln(1 + a_2) + \ln(1 + N)\}^{1/2} \leq \sqrt{\max(1, \ln(1 + a_2))\ln(1 + N)}.$$

Then, we prove when $N = 1/\epsilon^2 \times \ln^2(1 + 1/\epsilon^2)$.

$$\sqrt{1/N}\ln(1 + N) < \epsilon.$$

This is proved by

$$\sqrt{1/N}\ln(1 + N) \leq \epsilon \times \frac{\ln(1 + 1/\epsilon^2 \times \ln^2(1 + 1/\epsilon^2))}{\ln(1 + 1/\epsilon^2)}$$

$$\leq \epsilon \times \frac{\ln(1 + 1/\epsilon^2) + \ln(1 + \ln^2(1 + 1/\epsilon^2))}{\ln(1 + 1/\epsilon^2)}$$

$$\leq \epsilon + \epsilon \times \frac{\ln(1 + \ln^2(1 + 1/\epsilon^2))}{\ln(1 + 1/\epsilon^2)}$$

$$\leq \epsilon + \epsilon \times \frac{0.5\{1 + \ln^2(1 + 1/\epsilon^2))\}^{1/2} - 1}{\ln(1 + 1/\epsilon^2)}$$

$$\lesssim \epsilon.$$

From the third line to the fourth line, we use $\ln(x) \leq 0.5(x^{1/2} - 1)$ for $x > 0$. Then, the final statement is concluded. $\square$

# E   MORE COMPARISON TO XIE ET AL. (2021)

We briefly explain the guarantee when we use Algorithm 1 (Xie et al., 2021). For a given reward $r$, we first define a new feature class $\Phi_r^+$.

**Definition 23** (Augmented feature). *Let $\phi = [\phi_1, \cdots, \phi_d]$.*

$$\Phi_r^+ = \{\phi_r^+; \phi \in \Phi\}, \quad \phi_r^+ = [\phi_1, \cdots, \phi_d, r].$$

Then, we set

$$\mathcal{F} = \{a^\top \phi_r^+ \mid \|a\|_2 \le c\sqrt{d} + 1, \phi_r^+ \in \Phi_r^+\}.$$

where $c$ is some suitable constant. Given the hypothesis class $\mathcal{F}$ for the Q-function, we can run Algorithm 1 in (Xie et al., 2021). We denote the output policy as $\hat{\pi}$.

We check two assumptions to ensure the algorithm works. The first assumption is realizability. This is satisfied since for any policy $\pi \in \Pi$ ($\Pi$ is the class of all Markovian polices), we have $Q_{P^\star,r}^\pi \in \mathcal{F}$. The second assumption is completeness. This is also satisfied since $\mathcal{T}_{P^\star,r}^\pi \mathcal{F} \subset \mathcal{F}$ for any policy $\pi \in \Pi$ where $\mathcal{T}_{P^\star,r}^\pi$ is a Bellman-operator s.t.

$$\mathcal{T}_{P^\star,r}^\pi : \{\mathcal{S} \times \mathcal{A} \to \mathbb{R}\} \ni f \mapsto r(s,a) + \gamma \mathbb{E}_{s' \sim P^\star(s,a)}[f(s',\pi)] \in \{\mathcal{S} \times \mathcal{A} \to \mathbb{R}\},$$

where we denote $f(s,\pi) = \mathbb{E}_{a \sim \pi(s)} f(s,a)$. Then, by invoking their Corollary 5, we have

**Theorem 24** (PAC bound based on Xie et al. (2021)). *With probability $1 - \delta$,*

$$\forall \pi \in \Pi : V_{P^\star,r}^\pi - V_{P^\star,r}^{\hat{\pi}} \le c \frac{\sqrt{C_{\pi,r}^\dagger}}{(1-\gamma)^2} \left( \frac{(d+1)\log(1/\delta)\log|\mathcal{A}|}{n} \right)^{1/5}.$$

*where*

$$C_{\pi,r}^\dagger = \sup_{\phi_r^+ \in \Phi_r^+} \sup_{a \in \mathbb{R}^{d+1}} \frac{a^\top \mathbb{E}_{d_{P^\star}^\pi}[\phi_r^+ \{\phi_r^+\}^\top] a}{a^\top \mathbb{E}_\rho[\phi_r^+ \{\phi_r^+\}^\top] a}.$$

We compare the above result with our result in Theorem 6. First, since $C_r^\dagger$ includes $r$ and all possible features in $\Phi$, this partial coverage condition is stronger than ours (recall our partial coverage condition is only related to the true representation $\phi^\star$), and we always have $C_\pi^\star \le C_{\pi,r}^\dagger$. Secondly, the dependence on $n$ is much worse. Third, it is unclear whether the learned policy can compete against any history-dependent policy. Recall in Theorem 6, we show that our algorithm can compete with any history-dependent policies.

