# OpenReview forum: "Representation Learning for Online and Offline RL in Low-rank MDPs"
_ICLR.cc/2022/Conference — ICLR 2022 Spotlight_

### Official Review · Reviewer_SjqZ · 2021-10-19

**Correctness:** 3
**Technical Novelty And Significance:** 3
**Empirical Novelty And Significance:** Not applicable
**Recommendation:** 8
**Confidence:** 4

**Main Review:**

The almost optimism/pessimism seems not surprising to me. The key observation, as far as I understand, is that we can maintain the data distribution for the online setting via certain stop criteria in infinite horizon scenarios. That is to say, when we terminate the episode with probability $1-\gamma$ at each step, the state distribution we sample is exactly the stationary state distribution. If we uniform sample the action at the state we terminates, by standard generalization arguments (e.g. Lemma 12), we only need to pay an additional $|\mathcal{A}|$ factors, which will be fine. Also, such stationary distribution connects the value function with the sampling distribution of dataset, which is helpful for the theoretical derivation (e.g. in Lemma 9).

Although the overall idea is interesting, I have several concerns on this paper, both from the high-level perspective and the technical details.

1. If I understand correctly, the proposed algorithm can only be worked on the infinite horizon cases, as it heavily relies on the observation that when we terminate the episode with probability $1-\gamma$ at each step, we can return the sample of state following the stationary distribution. As far as I know, this does not hold for any real ''episodic’’ case, where we always finish at a fixed number of step. I would like to ask, is it possible to construct a similar algorithm for the episodic setting that terminates at a fixed number of step, following the idea proposed in the paper, as the authors argue they follow the exactly same setup as Flambe in Table 1, but the basic setting already differs. And if no such generalization, I will suggest the authors revising such claim and making the application of this trick and the potential limitation of this trick more explicitly.

2. I would also want to ask the question in the following technical way: how can we track the state visitation probability other than the policy cover methods in Flambe if we consider the fixed termination setting? I feel things can be even worse when the transitions can depend on the specific level $h\in[H]$ where $H$ is the length of each episode (which is the exact setting of Flambe). Does this paper provide new insights on that? Or is this paper only provide a cheap way for the policy cover, utilizing the structure of infinite horizon setting?

3. The proof part is not well organized and can be hard for the potential readers to understand. There are also some typos in the derivation, most of them are easy to fix. I want to name some typos that make me extremely confused at the first glance. At the end of Page 15, the authors claim to replace $\pi_n$ with $\bar{\pi}_n$ via importance sampling. However, $d$ is the stationary distribution, whose density ratio cannot be directly bounded by $|\mathcal{A}|$. Such transformation is needed for the application of ellipsoid potential lemma, and I guess here $d^{\bar{\pi}_n}$ is not the stationary distribution of $\bar{\pi}_n$, but the stationary distribution of $d^{\pi_n}(s) \times \text{Uniform}(a)$. Overall, the $\bar{\pi}_n$ is never defined. I hope the authors can go through the proof again to eliminate all of these ambiguities.

---------------
Update after the rebuttal: I check the proof again and understand the generalization to the time-inhomogeneous setting now. The one-step back trick can be conducted at any time step $h$ conditioned on the visiting distribution at $h-1$, and with recursion we can go back to the initial distribution $d_0$, which I now feel is the most significant contribution to the community. In this way, we don't need to construct point-wise upper bound like UCBVI and LSVI, which can be beneficial for the learning of low-rank MDP (as if I understand correctly, Flambe, in some sense, aims at providing point-wise upper bound so that they need to make the policy cover with the ellipsoid planner). I have adjusted my score accordingly.


**Summary Of The Paper:**

This paper consider the representation learning in low-rank MDPs under infinite-horizon settings. The authors provide the improved sample complexity and statistical error for the online and offline scenarios correspondingly, based on an observation called the almost optimism/pessimism at the initial state distribution.

**Summary Of The Review:**

The authors don’t properly address the difference between the newly proposed method and Flambe, and I feel the proposed method, although delicate, has so many restrictions that the authors don’t properly address. Meanwhile, the proof is not well-polished. I hope the authors can refine the writing during the rebuttal and make the intuition much more clear, compared with the current highlight of analysis in Section 4.2, that implicitly use several convenient property of the infinite horizon setting.

---

> ### Author Response · Authors · 2021-11-11
> **Rebuttal for reviewer SjqZ**
>
> Thank you for your detailed comments!
>
> Q: If I understand correctly, the proposed algorithm can only be worked on the infinite horizon cases, as it heavily relies on the observation....
>
> A:   We would like to start by claiming that our method can be easily modified for finite horizon time-inhomogeneous setting, while FLAMBE cannot be efficiently extended to infinite horizon discounted setting. Below, we give a detailed explanation.
>
> * Our function approximation setup and the computation oracle MLE are exactly the same as the FLAMBE’s. Regarding the horizon, it is not exactly the same. We did point out in the caption of the first table where we say that the horizon settings are different across these prior works. We will emphasize that more in the revised version.
>
> * Like many online RL algorithms such as PG-PG [Agarwal et.al, 20a]), we can easily modify Rep-UCB for the finite-horizon time-inhomogeneous setting as follows.
>
>   *  For each episode, we need to collect a tuple $ (s_t,a_t,s^{'}_t) $  over the horizons (a.k.a time steps) from $t=0$ to $ t=H$  in a forward manner.
>   *  Then, $s_t$ is sampled from $d^{\pi}_{P^{\star},t}(s)$. (marginal distribution at a time $t$ under $\pi$ ), $a_t$ is sampled from the uniform distribution over actions,  $s^{'}_t$ is sample from the unknown transition distribution,
>   *  MLE at the time step $t$ is performed on $D_{n,t}$ that is the historical data up to the episode $n$ at time $t$ ($1\leq t \leq H$),
>   *  Use the time-dependent bonus based on the learned feature to induce exploration.
>   *   Sample complexity-wise, we get almost the same dependence except that we will pay $H^4$ since we are operating in the time-inhomogeneous way now. This is still a significant improvement over FLAMBE (afterall,FLAMBE’s sample complexity cannot be magically reduced from $H^{22} /\epsilon^{10}$ to $H^4 / \epsilon^2$ simply by switching to the discounted infinite horizon setting.). This is a sharp contrast to the techniques used in FLAMBE.
>
> *  We deliberately decided to use the episodic infinite horizon time-homogeneous setting mainly to highlight the advantage of our method over FLAMBE. FLAMBE (as well as Moffle) cannot learn a stationary policy for episodic infinite-horizon time-homogeneous setting since their algorithms rely on a forward layer-by-layer exploration style approach. In our perspective, the ability to learn stationary policy is more relevant to practical considerations. Indeed classic policy gradient and policy optimization papers such as CPI [Langford and Kakade, 02], NPG [Agawal et.al, 21], and PC-PG [Agarwal et.al, 20a] all consider the episodic discounted infinite horizon setting.
>
> Finally,  we want to highlight how we can handle the finite-horizon time-inhomogeneous setting theory-wise.  A key Lemma 13 (one-step-back inequality) is currently used to prove Lemma 9. It is stated for the stationary distribution $d^{\pi}_{P^{*}}$ and the offline data $\rho_n$.
>
>  In the finite horizon case, lemma 13 still holds by using the marginal distribution at a time-step $t$. More going into details,
>
> *  $d^{\pi}_{P^{*}}$ in the left-hand side is replaced with
>
> $d^{\pi}_{P^{*},t}$ (marginal distribution at a time-step t, 1<=t<=H) .
>
> * $d^{\pi}_{P^{*}}$ in the right-hand side is replaced  with
>
> $d^{\pi}_{P^{*},t-1}$ (marginal distribution at t-1).
>
> * $\rho_n$ in the right hand size is replaced by  $\rho_{n,t}$.
>
> * Then, we can operate the one-step-back trick for each time step.
>
> * Finally, it is concluded that we can still apply an elliptical potential lemma by using the feature $\phi^{*}_t(s,a)$ for $1\leq t\leq H$; then, Lemma 9 is similarly concluded.
>
> Q: I would also want to ask the question in the following technical way: how can we track the state visitation probability other than the policy cover methods in FLAMBE if we consider the fixed termination setting?....
>
> A: We do not need to track the state visitation here as we are not doing reward-free exploration. Like UCB style approach (e.g., UCB-VI), our approach combines exploration and exploitation and uses optimism, and directly aims for learning a single stationary policy that is near optimal. Unlike FLAMBE, we do not need to explore in a layer-by-layer fashion or keep tracking whether or not we have explored enough in the previous layers. In other words, we completely abandoned the layer-by-layer forward exploration algorithmic framework from FLAMBE (and others including Moffle, etc.). Like standard UCB approaches,  our algorithm, as we explained above, easily works for finite-horizon setting as well.
>
> Q: The proof part is not well organized and can be hard for potential readers to understand.....
>
> A: We apologize for causing the confusion. We will clean up the appendix to make it more readable. For $d^{\bar \pi_n}$, we defined it at the beginning of Appendix A. And this is equivalent to the quantity as you guessed. We will recall this definition when we use the notation.

---

> > ### Comment · Reviewer_SjqZ · 2021-11-11
> > **Thanks for the clarification. I have adjusted the score accordingly.**
> >
> > Thanks for the quick response. I have adjusted the score after I check the proof strategy on time-inhomogeneous cases provided in the rebuttal. I would like to suggest the authors providing much more discussions on that in the revision, and emphasizing more on the one-step back trick, which I think is the most significant part in this paper.
> >
> > Also, I'm still a little confused on the improvement over Flambe. I would like to ask if my understanding in the update of review is correct. That is to say, Flambe still in some sense aimed at point-wise upper bound, while the REP-UCB avoids that. Hence REP-UCB don't need to construct policy cover to enforce the exploration, which is instead guaranteed by the optimism in the initial distribution. Thanks in advance.

---

> > > ### Author Response · Authors · 2021-11-11
> > > **Thank you for your quick reply!**
> > >
> > > Thank you for such a quick reply!
> > >
> > > * Yes, we will discuss the finite-horizon time-inhomogeneous setting, and how the one-step-back trick works in the next version.
> > >
> > > * To the best of our knowledge, Flambe actually does not use the idea of optimism in the face of uncertainty, since it does not aim to find a near-optimal policy directly, instead, it aims to find a set of policy-covers where each policy-cover is responsible for fully exploring a particular layer h.  Rep-UCB interleaves exploration and exploitation via the traditional principle of optimism in the face of uncertainty. I.e., Rep-UCB does not waste samples for over exploration once it finds a near-optimal policy (e.g., consider the setting where reward function is dense and informative, there is no need for us to explore everywhere)

---

> > > > ### Comment · Reviewer_SjqZ · 2021-11-11
> > > > **Thanks for the clarification.**
> > > >
> > > > Thanks for the clarification again. I would like to rephrase the question I would like to ask.
> > > >
> > > > I understand that Flambe and Moffle are not optimism-based. But in my understanding, the ellipsoid planner used in these works aimed at finding the direction that the data do not cover on the covariance matrix induced by the learned representation. This is in some sense similar to the UCB you use here. And I feel in Flambe, the authors used the policy cover, mainly due to the reason that the representation itself introduces additional error, and hence we cannot guarantee point-wise upper bound for the $Q$ function if we view the term $\\|\hat{\phi}(s, a)\\|_{\hat{\Sigma}}$ as a bonus term. I would like to know if that understanding makes sense.
> > > >
> > > > Or in other words, I would like to ask, if it's possible to generalize the idea used in this work to the UCB-based reward-free exploration, e.g. [1]. I understand this is out of the scope of the paper, but I feel such discussion can benefit the understanding on the significance of the result.
> > > >
> > > > I may have some misunderstanding on the motivation of policy cover used in Flambe, so perhaps my question is a little bit silly. Thanks for the clarification in advance.
> > > >
> > > > [1] Kaufmann, Emilie, Pierre Ménard, Omar Darwiche Domingues, Anders Jonsson, Edouard Leurent, and Michal Valko. "Adaptive reward-free exploration." ALT 2021

---

> > > > > ### Author Response · Authors · 2021-11-11
> > > > > **Further discussion**
> > > > >
> > > > > What you described for FLAMBE makes sense to us, indeed, yes, due to the error in the learned representation and the fact that it is changing over iterations, it cannot be used to provide point-wise optimism.
> > > > >
> > > > > Regarding reward free exploration, we think Rep-UCB can do it. We thought about this question after the submission, but we have not get chance to verify things carefully. Intuitively, when we just run Rep-UCB with our reward bonus, we think we can still use the one-step back trick to track the elliptical potential formed under the true representation $\phi^\star$. We will finish exploration if we have explored all possible directions in the $\phi^\star$ space. In this case, Rep-UCB will maintain a policy cover as well. But in time-homogeneous case, we will just learn a single policy-cover, while we believe Flambe still has to learn a set of policy-covers (i.e., flambe will have to convert infinite horizon case to finite case with $H = \Theta(1/(1-\gamma))$, and run its layer-by-layer exploration algorithm).
> > > > >
> > > > > We think even under the reward free exploration case, we can still reduce the sample complexity of FLAMBE by a margin. Somehow flambe is using older exploration technique -- absorbing MDPs, originated from $E^3$, to perform exploration, which we know cannot give us $1/\epsilon^2$ sample complexity.
> > > > >
> > > > > Said though, we cannot affirmatively say here the Rep-UCB can work --- we will verify it.

---

### Official Review · Reviewer_99V8 · 2021-10-31

**Correctness:** 3
**Technical Novelty And Significance:** 2
**Empirical Novelty And Significance:** 2
**Recommendation:** 5
**Confidence:** 4

**Main Review:**

The paper is well written.

The paper provides a few tangible novel definitions such as partial coverage in offline RL.

However, the paper needs further work to be ready.

1- The first major drawback of this paper is that the online algorithm just uses one sample per episode, no matter how long is the episode. I doubt anyone would implement this algorithm in practice and the insight is limited. I understand it makes the theory cleaner, but the authors are encouraged to provide a full study.
Also, it is not clear how this algorithm might be extended to undiscounted finite-horizon MDPs, a subset of episodic MDPs.

2- It seems the work assumes M is finite. While extensin to infinite set might be straightforward, including it in the paper is necessary. In that case, is the MLE part efficient?
Note that is it quite a limiting setting since state-space seems to be large.

3-  The condition that for any mu and phi \int_s' mu(s') phi(s,a)= 1 is very strong and very limiting. The authors are encouraged to relax it.


I might have been wrong in the above statements.

The paper studies an important problem, but some details are still left to be developed and the authors are encouraged to address them.




**Summary Of The Paper:**

This paper studies low-rank episodic MDPs when the reward is deterministic and the reward function is known. The paper proposes a method that collects data, uses the data to estimate the low-rank MDP, uses this estimate, confidence bound around it, to come up with the policy to be used for the next episode.

The algorithm provides PAC style bound on how many episodes are needed to learn epsilon optimal policy.

The authors extend the scope of this work to offline low-rank MDP where a logged data set is given where an RL algorithm is needed to learn and optimize to come up with a new policy.








**Summary Of The Review:**

The paper is well written,
Address an important problem.
However, there are pieces of this work that are still missing to complete the work.

---

> ### Author Response · Authors · 2021-11-11
> **Rebuttal for reviewer  99V8**
>
> Thank you very much for your encouraging comments.
>
> Q: The first major drawback of this paper is that the online algorithm just uses one sample per episode, no matter how long is the episode. I doubt anyone would implement this algorithm in practice and the insight is limited.
>
> A:  Though we can see your concern, we would not see it that way since the same concern would be applied to classical landmarking algorithms such as Policy Gradient [Sutton et.al, 2000], CPI [Kakade and Langford, 2002], and NPG (natural policy gradinet) [Agarwal et.al, 2021] . For example, the policy gradient theorem (Theorem 1 in [Sutton et.al, 200]) for the infinite discounted horizon setting states that the sampling distribution needs to be a discounted occupancy distribution, i.e., one sample per episode. This kind of sampling oracle is also assumed in NPG’s analysis (please refer to Algorithm 1 in NPG  [Agarwal et.al, 2021], which is the same sample oracle we used), and CPI's analysis. In practice, one may not throw away all samples to gain some sample efficiency at the cost of not having i.i.d samples. However, this does not mean that policy gradient, NPG, and CPI are useless.
>
> Sutton, R. S. et.al (2000). Policy gradient methods for reinforcement learning with function approximation. In Advances in neural information processing systems (pp. 1057-1063).
>
> Kakade, S., & Langford, J. (2002). Approximately optimal approximate reinforcement learning. In Proc. 19th International Conference on Machine Learning.
>
> Agarwal, A. et.al (2021). On the theory of policy gradient methods: Optimality, approximation, and distribution shift. Journal of Machine Learning Research, 22(98), 1-76.
>
> Q:  I understand it makes the theory cleaner, but the authors are encouraged to provide a full study. Also, it is not clear how this algorithm might be extended to undiscounted finite-horizon MDPs, a subset of episodic MDPs.
>
> A:   This is possible. We answer this question in the rebuttal for MSqC,
>
> Q: It seems the work assumes M is finite. While extensin to infinite set might be straightforward, including it in the paper is necessary. In that case, is the MLE part efficient? Note that is it quite a limiting setting since state-space seems to be large.
>
> A: That’s a good point. By following the convention of the RL theory papers with general function approximation (e.g., Bellman rank [1], Witness rank [2],  FLAMBE, Homer and classic contextual bandit papers), we only focus on the case where M is finite. Note that our sample complexity depends on the log(M), which is a very standard statistical complexity of the hypothesis class.
>
> While our function classes are general (i.e., we do not assume specific parameterizations),  when we specialize them to specific parameterization, we will be able to extend the MLE generalization to an infinite hypothesis class. Note that all we require is the MLE generalization bound, which is well studied in statistics for different model classes including continuous classes (e.g., [Geer, S. A., 2000]).
>
> In practice,  the extension to the infinite hypothesis class is also computationally feasible since the MLE part is just a standard supervised learning problem with log-loss. In practice, one may parameterize these models using neural networks and optimize them via gradient descent approaches (e.g., Kaiser et.al 2020). This reduction framework is also widely adopted in recent works of RL with general function approximation including the closest prior work FLAMBE and papers in bandits [Dudik et al., 2017].
>
> Kaiser, L. et al. (2020). Model-based reinforcement learning for atari. arXiv preprint arXiv:1903.00374.
>
> Jiang, N., et al. (2017) Contextual decision processes with low Bellman rank are PAC-learnable. In International Conference on Machine Learning (pp. 1704-1713). PMLR.
>
> Sun, W. et al. (2019). Model-based RL in contextual decision processes: Pac bounds and exponential improvements over model-free approaches. In Conference on learning theory (pp. 2898-2933). PMLR.
>
> Geer, S. A., & van de Geer, S. (2000). Empirical Processes in M-estimation (Vol. 6). Cambridge university press.
>
> Q:The condition that for any mu and phi \int_s' mu(s') phi(s,a)= 1 is very strong and very limiting. The authors are encouraged to relax it.
>
> A: While we can see your concern, this can be easily relaxed. We answer this question in the rebuttal for MSqC,

---

### Official Review · Reviewer_MSqC · 2021-11-01

**Correctness:** 4
**Technical Novelty And Significance:** 3
**Empirical Novelty And Significance:** Not applicable
**Recommendation:** 6
**Confidence:** 4

**Main Review:**

Strengths:

This paper improves the sample complexity bound of [FLAMBE](https://arxiv.org/abs/2006.10814) by doing the exploration, exploitation, and model selection at the same time. This interplay is much more practical and interesting than the previous explore and commit version. Also, in the offline algorithm (Rep-LCB), the proposed relative condition number provides some insight into the coverage of the offline dataset. It also makes sense by comparing the result of the offline algorithm with the policy covered by the offline dataset.

Weakness:

First, I'm concerned with the actual sample complexity of the algorithm. The authors calculate the sample complexity by counting $N$, which is the total episodes. However, it takes $poly((1 - \gamma)^{-1})$ in expectation to sample state $s$ from distribution $d_{P^*}^{\pi}$. Also, [FLAMBE](https://arxiv.org/abs/2006.10814) considered a time-inhomogeneous model for finite-time horizon setting, while here a time-homogeneous, infinite-time horizon setting is considered. I wonder if it is fair to directly convert the $H^{22}$ to $(1 - \gamma)^{22}$ here.

Second, the realizability assumption (Assumption 2) sounds wired to me. Why we need **any** combination of $\mu$ and $\phi$ is a (signed) measure (i.e. $\int_{s'} \mu^\top(s')\phi(s, a)\mathrm d(s') = 1, \forall \mu, \phi, s, a$)? This assumption sounds too restrictive to me, and it has never appeared in the previous literature (like [FLAMBE](https://arxiv.org/abs/2006.10814)) to the best of my knowledge. It would be better if the authors can provide some examples satisfying that assumption.

Third, though the previous literature made the same Maximum Likelihood Oracle as in Definition 3 here, I'm wondering if this optimization oracle can be solved efficiently for an arbitrarily large state space. It would be beneficial if the authors can provide some optimization solutions for this oracle.

**Summary Of The Paper:**

This paper studies the representation learning for linear MDP. The authors provide an online algorithm and its offline counterpart. Theoretical analysis justifies that the proposed algorithms are sample efficient.

**Summary Of The Review:**

This paper studies the representation learning in linear MDP and significantly improves the previous sample complexity. However, as mentioned above, I'm concerned with the following aspects:
- Sample complexity of the algorithm and fairness comparing two algorithms in different settings (time-inhomogeneous v.s. time-homogeneous)
- Assumption 2 sounds too restrictive for me.
- Definition 3, though appeared in the literature, I wonder if it can be solved efficiently

Based on the concerns above, I lean towards rejecting this paper but I'm open to further discussions.

-- post rebuttal
I've updated my score since the authors clearly addressed my concerns.

---

> ### Author Response · Authors · 2021-11-11
> **Rebuttal for reviewer MSqC**
>
> Thank you very much for your feedback!
>
> Q: Sample complexity of the algorithm and fairness comparing two algorithms in different settings (time-inhomogeneous v.s. time-homogeneous)
>
> Thanks for pointing out the difference. Being aware of that, we also pointed out at the end of the caption of table 1, the horizon dependence is not directly comparable.  But we argue our comparison over FLAMBE is still fair since we can easily extend our algorithm to finite horizon setting. We deliberately decided to use the infinite horizon time-homogeneous setting mainly to highlight the difference of our method from FLAMBE. Below we give the detailed explanations.
>
> * Like many UCB-style RL algorithms, we can easily modify Rep-UCB for the finite-horizon time-inhomogeneous setting. Concretely, we need to change the algorithm as follows:
>
>   *  For each episode, we need to collect a tuple $ (s_t,a_t,s^{'}_t) $  over the horizons (a.k.a time steps) from $t=0$ to $ t=H$  in a forward manner.
>   *  Then, $s_t$ is sampled from $d^{\pi}_{P^{\star},t}(s)$. (marginal distribution at a time $t$ under $\pi$ ), $a_t$ is sampled from the uniform distribution over actions,  $s^{'}_t$ is sample from the unknown transition distribution,
>   *  MLE at the time step $t$ is performed on $D_{n,t}$ that is the historical data up to the episode $n$ at time $t$ ($1\leq t \leq H$),
>   *  Use the time-dependent bonus based on the learned feature to induce exploration.
>   *   Sample complexity-wise, we get almost the exact same dependence except that we will pay $H^4$ since we are operating in the time-inhomogeneous way now. This is still a significant improvement over FLAMBE (afterall,FLAMBE’s sample complexity cannot be magically reduced from $H^{22} /\epsilon^{10}$ to $H^4 / \epsilon^2$ simply by switching to the discounted infinite horizon setting.). This is a sharp contrast to the techniques used in FLAMBE.
> * FLAMBE would be not able to to solve the infinite-horizon time-homogeneous setting efficiently. While we can naively set H in the order of $1/(1-\gamma)$ when moving from finite horizon to infinite discounted setting, FLAMBE will perform its forward layer-by-layer style exploration.  Thus, FLAMBE cannot learn a stationary policy.  Compared to FLAMBE, the exploration and exploitation in REP-UCB in the infinite-horizon time-homogeneous setting is simultaneously performed over time steps and we only aim to learn a stationary policy that is near optimal.
> * In our perspectives, the episodic infinite horizon discounted setting is more relevant to practice as this setting is used in papers such as CPI and PG.  We mention this point in the answer for SjqZ in detail.
>
> Q: Assumption 2 sounds too restrictive.
>
> A:  We would like to mention three points (1) this is assumed in FLAMBE, (2) latent variable model satisfies this assumption, (3) it can be easily relaxed. We will add this discussion in the revision.
>
> * Assumption 2 is required for FLAMBE to work. We have confirmed this point with FLAMBE’s authors via emails. In particular, while it has not been explicitly written out in the FLAMBE paper, FLAMBE uses MLE and model-based planner in their algorithm, they do require all $(\mu,\phi)$ pairs to form a **valid transition kernel**. Otherwise, the MLE generalization bound won’t go through.
>
> * This is a standard assumption in many model-based RL works. In our case, one choice of parameterizations is to use the latent variable model where $\phi$ will be a mapping from state action space to d-dim simplex, and $\mu$ will be parameterized such that each column of $\mu$ is a distribution over state space. This way, $(\phi,\mu)$ pair gives a **valid transition kernel**. Indeed, even when states are images, in practice, it is not too hard to parameterize models that give valid transition kernels (e.g., Kaiser et.al 2020 constructs models which are condition distributions over image space and optimizes models via MLE using log-loss).
>
> * This assumption can be easily relaxed. We can just require the normalizing constant $\int_{s’} \mu^{\top}(s’)\phi(s,a)$ to be positive. Then, we can modify the MLE procedure to convert the original class into the normalized class by performing the normalization $\mu^{\top}(s’)\phi(s,a)/ \int_{s’} \mu^{\top}(s’)\phi(s,a)$. The MLE generalization lemma still goes through. All other arguments still go through with this modified class. The current assumption  $\int_{s’} \mu^{\top}(s’)\phi(s,a)=1$ is just imposed to simplify the presentation.
>
> Kaiser et. al (2020). Model-based reinforcement learning for atari.
>
> Q: Definition 3, though appeared in the literature, I wonder if MLE partcan be solved efficiently
>
> A: We also acknowledge that the MLE can be nonconvex optimization and we mainly treat this as a black-box supervised learning oracle. This is still practically feasible. In practice, one may parameterize these models using neural networks and optimize them via gradient descent (e.g., Kaiser et.al 2020).

---

> > ### Comment · Reviewer_MSqC · 2021-11-12
> > **Thank you for your response!**
> >
> > The authors addressed my previous concern regarding the comparison vs. FLAMBE. For the too restrictive assumption, the authors provided methods to relax that. Given that I will raise my score.

---

### Official Review · Reviewer_cvhG · 2021-11-02

**Correctness:** 4
**Technical Novelty And Significance:** 4
**Empirical Novelty And Significance:** Not applicable
**Recommendation:** 8
**Confidence:** 4

**Main Review:**

This paper is well-motivated: representation learning is a very important question in RL, both empirically and theoretically. This paper is also well-written, and the technical part of this paper is easy-to-follow.

On the plus side:
-	The REP-UCB algorithm is conceptually very simple: it alternates between learning a model (using MLE) and finding an optimistic policy. In terms of the computation complexity, the REP-UCB algorithm is oracle efficient, using the same computation oracle as FLAMBE.
-	This paper significantly reduces the sota sample complexity in the online setting to a reasonable level. Although technically speaking, FLAMBE focuses on reward-free exploration, where this paper focus on a standard online setting.
-	For the offline setting, this paper achieves polynomial sample complexity with partial coverage, where the coverage is measured by the ground-truth feature. In contrast, previous results in the offline setting assumes a known representation, or requires a coverage over all hypothesis.

On the minus side, I have the following questions/concerns:
-	The sample complexity in this paper is measured by the number of episodes N. In each episode, the algorithm collects one (s,a,s’) tuple from distribution d^{\pi}_{P^*} with uniformly random action. However, after taking a random action, the distribution of the state changes. Does it mean that, in order to collect one transition tuple, the algorithm needs to interact with the environment for multiple steps? How does this mechanism affect the sample complexity?
-	-	In the proof of Lemma 12, the last equation block in Page 18: Could you elaborate on the first inequality (the MLE guarantee)? If I understand correctly, the inequality is doing a change of measure (from \hat{P} to P^\star). But the MLE guarantee only upper bounds the expected one norm squared. So a naïve Jensen gives an $\sqrt{\zeta_n}$ upper bound, instead of \zeta_n stated in the paper. Then $n\sqrt{\zeta_n}$ is on the order of \sqrt{n}, and I expect that this bound is too loose for later steps.

Additional questions/remarks:
-	Did the authors try regret minimization in the online setting? It seems to me that the random action in data collection creates some technical difficulty.
-	In Page 14, the second displayed equation in the proof of Lemma 8: it should be ${\hat{\Sigma}_{n,\phi}^{-1}}$.

======= after rebuttal =======
The authors' response addressed my concerns. I'll raise my score according.

**Summary Of The Paper:**

This paper studies representation learning in low rank MDPs in both online and offline setting. This paper proposes REP-UCB algorithm, which improves the sample complexity of FLAMBE significantly. In addition, REP-UCB is much simpler than FLAMBE.  For the offline setting, the REP-LCB algorithm also achieves polynomial sample complexity with partial coverage. In particular, the coverage is measured by the ground-truth feature.

**Summary Of The Review:**

This paper is well motivated, well written, and the results are impressive. But I don’t fully understand some of the derivations (see main review). So at this point, I can only suggest a weak accept with low confidence. If my concerns were addressed, I’m happy to raise my score.

---

> ### Author Response · Authors · 2021-11-11
> **Rebuttal for reviewer cvhG**
>
> Thank you for your positive and incisive comments.
>
> Q: The sample complexity in this paper is measured by the number of episodes N. In each episode, the algorithm collects one (s,a,s’) tuple from the distribution $d^{\pi}_{P^*}$ with uniformly random action. However, after taking a random action, the distribution of the state changes. Does it mean that, in order to collect one transition tuple, the algorithm needs to interact with the environment for multiple steps? How does this mechanism affect the sample complexity?
>
> A: Yes, it is! To collect one ($s$) from $d^{\pi}_{P^*}(s)$, we need at most $O(1/(1-\gamma))$ interactions with high probability.   Then, the total sample complexity changes from  $1/(1-\gamma)^2$ to  $1/(1-\gamma)^3$. After the initial submission, we have edited the result such that all reported sample complexities are measured by the number of interactions. Please refer to the edited version.
>
> Q: In the proof of Lemma 12, the last equation block in Page 18: Could you elaborate on the first inequality (the MLE guarantee)?
>
> A: We appreciate that you looked into the details of the proof and checked this point. After the initial submission, we noticed and have already fixed them in the new version.  We can still keep $\zeta_n$ without deteriorating the rate like $\sqrt{\zeta_n}$.
>
> The argument goes through as follows. It stars from
> $ [\int \sum_a \hat \mu_n(s)\pi(a\mid s) g(s,a)d(s)] ^{\top}[n E_{(s,a)\sim \rho_n}[\hat \phi_n\hat \phi^{\top}_n]+\lambda I ][\int \sum_a \hat \mu_n(s)\pi(a\mid s) g(s,a)d(s)]. $
>
> By using the norm assumption, this is upper bounded by
> $$ n E_{(\tilde s, \tilde a)\sim \rho_n}\{[\int \sum_a \hat \mu^{\top}_n (s)\hat \phi_n(\tilde s,\tilde a)\pi(a\mid s) g(s,a) d(s)]^2 \} + B^2 \lambda_n d .$$
>
> This is equal to
> $n E_{(\tilde s,\tilde a) \sim \rho_n}[ E_{s \sim \hat P(\tilde s,\tilde a), a \sim \pi(s) }[g(s,a)]^2]+ B^2 \lambda_n d$.
>
> We use MLE guarantee. Then, the above is upper bounded by
>
> $n E_{(\tilde s,\tilde a) \sim \rho_n}[ E_{s \sim P^{\star}(\tilde s,\tilde a), a \sim \pi(s) }[g(s,a)]^2]+ B^2 \lambda_n d + n \zeta_n B^2 $
>
> Finally, we use Jensen’s inequality. The above is upper-bounded by
>
> $ n E_{(\tilde s,\tilde a) \sim \rho_n}[ E_{s \sim P^{\star}(\tilde s,\tilde a), a \sim \pi(s) }[g^2(s,a)]]+  B^2 \lambda_n d + n \zeta_n B^2.$
>
> The rest of the argument goes through following the previous derivation.
>
>
> Q: Did the authors try regret minimization in the online setting?
>
> A: This is an exciting open problem. Uniform exploration in the action space, which prevents the regret minimization, seems to be necessary in the currently known algorithmic framework for representation learning, and is also a key component in FLAMBE and Moffle. At this point, we have not figured out how to extend our work to the regret minimization. But this is something that we always aim towards in ongoing/future research. In particular, we think we can potentially apply techniques in Dong et.al 2020 below.
>
> Dong, K., Peng, J., Wang, Y., & Zhou, Y. (2020, July). Root-n-regret for learning in markov decision processes with function approximation and low bellman rank. In Conference on Learning Theory (pp. 1554-1557). PMLR.
>
> Q: In Page 14, there are typos.
>
> Thank you for catching typos! We have fixed them.

---

> > ### Comment · Reviewer_cvhG · 2021-11-11
> > **Thanks for the quick response!**
> >
> > Thanks for the quick response! My concerns are addressed, and I'll raise my score accordingly.
> >
> > Minor typo: in the proof of Lemma 12, the equality in line (MLE guarantee) should be inequality.

---

### Author Response · Authors · 2021-11-11
**Summary of Rebuttal**

We would like to thank the reviewers for their comments. In summary, reviewers mainly bring up the concern over (1) whether it is fair to compare with FLAMBE given our setting is time-homogeneous and FLAMBE’s setting is finite-horizon time-inhomogeneous, (2) the assumption of low-rank MDPs (every $\hat \mu^{\top}\phi$ needs to be a valid transition kernel).

We have addressed all of the points: (1) REP-UCB can be easily extended to the finite horizon time-inhomogeneous setting, while FLAMBE indeed cannot be efficiently extended to the infinite-horizon time-homogeneous setting. We deliberately chose the infinite-horizon setting to emphasize this difference.,  (2) it is also assumed in FLAMBE and it can be easily relaxed.

---

### Decision · Program_Chairs · 2022-01-20

**Decision:**

Accept (Spotlight)

**Comment:**

In this paper, the authors extend the FLAMBE to the infinite-horizon MDP and largely improved the sample complexity of the representation learning in FLAMBE. Meanwhile, the authors also consider the offline representation learning with the same framework. Although there is still some computational issue in MLE for the linear MDP, the paper completes a solid step towards making linear MDP for practice. The paper could be impactful for the RL community.

As the reviewers suggested, there are still several minors to be addressed:

- The extension of the proposed algorithm for finite-horizon MDP should be added.
- The directly comparison between the sample complexity of FLAMBE and the proposed algorithm in infinite-horizon MDP is not appropriate. The authors should clarify the difference here.
- The organization of the proof is not clear. As reviewer suggested, the one-step back trick should be emphasized for better significance of the submission.